# Cross-population GWAS and proteomics improve risk prediction and reveal mechanisms in atrial fibrillation

Shuai Yuan [1,2,3,4,16] ✉, Jie Chen [5,6,16], Xixin Ruan [5], Yuying Li [3], Sarah A. Abramowitz[1], Lijuan Wang [7], Fangyuan Jiang[7], Ying Xiong [3], Michael G. Levin [4,8], Benjamin F. Voight [4,9,10,11], Dipender Gill [12], Stephen Burgess [13,14], Agneta Åkesson [2], Karl Michaëlsson[15], Xue Li [7], Scott M. Damrauer [1,4,9,17] & Susanna C. Larsson [2,15,17]

Atrial fibrillation (AF) is a common cardiac arrhythmia with strong genetic components, yet its underlying molecular mechanisms and potential therapeutic targets remain incompletely understood. We conducted a cross-population genome-wide meta-analysis of 252,438 AF cases and identified 525 loci that met genome-wide significance. Two loci of *PITX2* and *ZFHX3* genes were identified as shared across populations of different ancestries. Comprehensive gene prioritization approaches reinforced the role of muscle development and heart contraction while also uncovering additional pathways, including cellular response to transforming growth factor-beta. Population-specific genetic correlations uncovered common and unique circulatory comorbidities between Europeans and Africans. Mendelian randomization identified modifiable risk factors and circulating proteins, informing disease prevention and drug development. Integrating genomic data from this cross-population genome-wide meta-analysis with proteomic profiling significantly enhanced AF risk prediction. This study advances our understanding of the genetic etiology of AF while also enhancing risk prediction, prevention strategies, and therapeutic development.

Atrial fibrillation (AF) is the most common arrhythmia, characterized by disorganized atrial depolarizations, which can lead to symptoms including palpitations and decreased exercise capacity, as well as more serious complications. With an aging global population, AF has become an epidemic and important health issue with increasing incidence and prevalence[1], particularly in North America and Europe[2]. The Global Burden of Disease 2019 Study estimated that approximately 59.7 million individuals live with AF, which is associated with 8.4 million disability-adjusted life years worldwide[3]. Hence, there is an urgent need to elucidate the etiological basis of AF to improve risk prediction, prevention and treatment.

While environmental factors play a role in AF development, the genetic contribution to AF susceptibility has been increasingly recognized. Multiple genome-wide association studies (GWASs) have uncovered over 100 risk loci, shedding light on AF's genetic architecture[4–8]. However, existing studies have largely been conducted in European populations, and a larger, more diverse GWAS—particularly one including multi-populations—could enhance the discovery of variants with smaller effects, as well as population-specific and shared loci.

Genomic data are now widely leveraged to improve disease risk prediction, identify risk factors, and facilitate therapeutic development. While some studies have explored these aspects for AF,

integrating cross-population genetic data with proteomic insights in a large-scale study could further refine the identification of genetic signals, associated comorbidities, causal risk factors, and potential drug targets. Thus, we conducted a cross-population GWAS meta-analysis involving over 2 million individuals and performed comprehensive downstream analyses to uncover unreported genetic loci, identify causal risk factors, and enhance AF risk prediction and therapeutic opportunities.

## Results

### Cross-population GWAS meta-analysis identified 379 unreported loci

We conducted a cross-population GWAS meta-analyses and a series of downstream analyses on AF (Fig. 1). The European meta-analysis, which included 228,926 AF cases from nine studies (Supplementary Data 1), identified 493 genetic loci reaching genome-wide significance (Supplementary Data 2). Among these, five loci showed significant evidence of heterogeneity in effect estimates across the contributing GWAS (heterogeneity test; $P < 0.05/493$, Supplementary Data 2). Of the 493 loci, 479 displayed consistent effect estimates between the discovery and replication datasets (Supplementary Data 2 and Supplementary Fig. 1), and 426 had $P < 0.05$ in the replication study (Supplementary Data 2). Using linkage disequilibrium score regression (LDSC) with the 1000 Genomes European reference panel, common variants explain 11.2% (95% CI: 9.2%–13.2%) of the variance in AF liability, assuming a 2% disease prevalence.

The cross-population GWAS meta-analysis, which included 252,438 AF cases, identified 525 loci that met genome-wide significance (Fig. 2a). Thirteen loci demonstrated significant heterogeneity in effect estimates ($P_{het} < 0.05/525$, Supplementary Data 3). The majority of risk alleles conferred small-to-moderate effect sizes, with odds ratios (ORs) ranging from 1.0 to 1.3 per allele (Fig. 2b). However, six lead SNPs had ORs exceeding 1.3 and were located in loci with genes *SORCS3, POLD1, AGBL4, AC126283.1, PITX2,* and *FAM241A* (Fig. 2b). Among the 525 significant loci, the breakdown by population revealed 483 loci in Europeans, 29 in East Asians, 5 in Africans, and 2 in Admixed Americans (Fig. 2c). Two loci of *PITX2* and *ZFHX3* genes were identified as shared across these populations (Fig. 2d).

### Comprehensive gene prioritization refined pathway exploration

Using a systematic prioritization framework, we nominated a likely causal gene at each of the 504 genome-wide significant loci, acknowledging that this assignment is based on available functional evidence and may not be definitive for all loci. Among these, 70 genes harbored protein-altering variants, and 47% of prioritized genes had ≥ 80% agreement across available methods (Supplementary Data 4).

To gain mechanistic insights into AF, we performed pathway enrichment analysis using these 504 prioritized genes. Enrichment analysis in the Reactome database identified 5 out of 1131 pathways significantly associated with AF after Bonferroni correction. Among these, muscle contraction and cardiogenesis showed strong associations (Fig. 3a and Supplementary Data 5). In addition, we conducted enrichment analysis using the Gene Ontology (GO) database. After Bonferroni correction, we identified 50 biological processes (BP), 7 cellular components (CC), and 6 molecular functions (MF) (Fig. 3b and Supplementary Data 6). GO enrichment analysis reinforced the role of

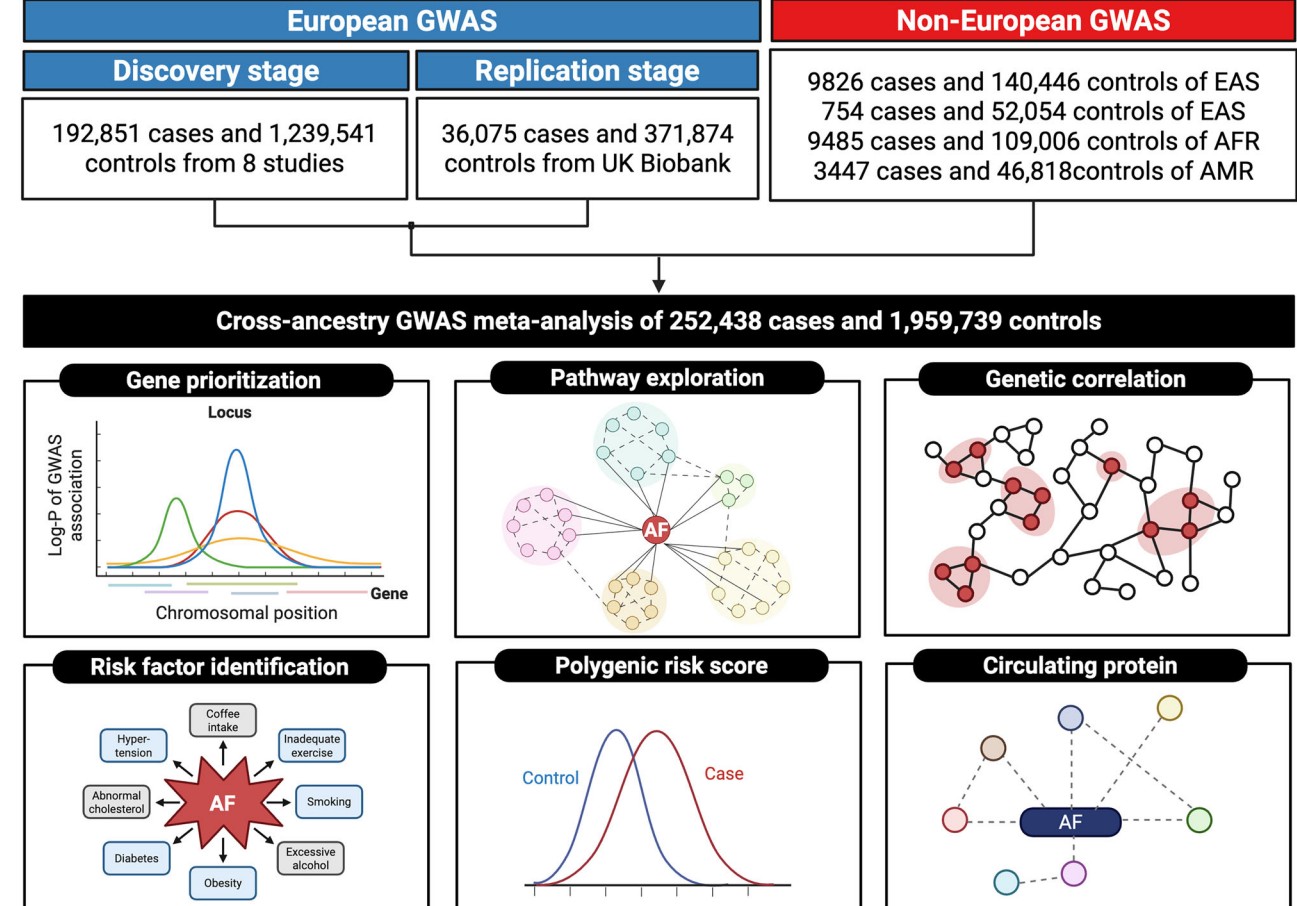

**Fig. 1 | Study design overview.** AF atrial fibrillation, AFR African, AMR Admixed American, GWAS genome-wide association study. EAS Eastern Asian, EUR European, SAS South Asian.

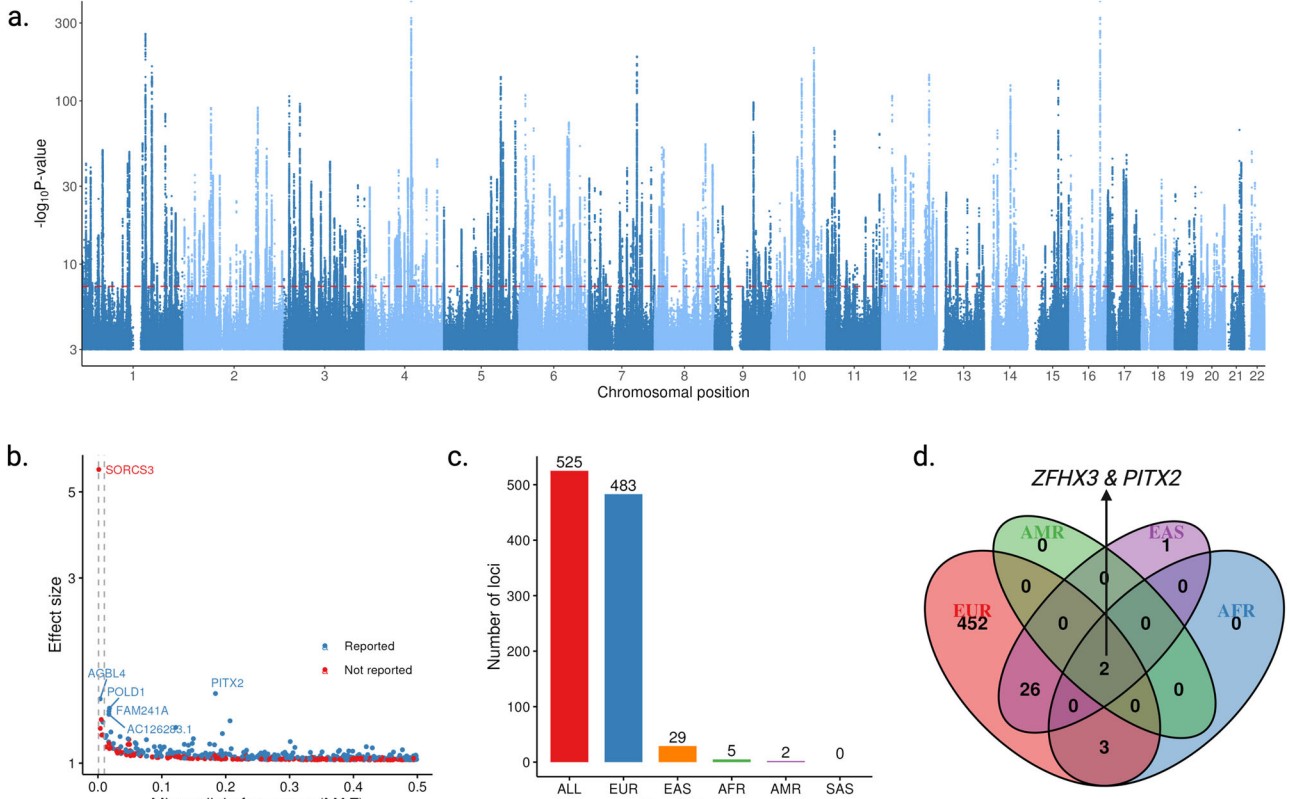

**Fig. 2 | Genetic loci associated with atrial fibrillation (AF) across populations of different ancestries. a** Manhattan plot of GWAS associations. The x-axis represents the genomic positions of SNPs across chromosomes, while the y-axis displays the -log10($P$) values, indicating the strength of the association. Each dot represents a single SNP, positioned based on its genomic location and statistical significance. The red dashed line marks the genome-wide significance threshold ($P = 5 \times 10^{-8}$). The statistical test was two-sided, and the Bonferroni-corrected significance level was applied. **b** Scatter plot of minor allele frequency (MAF) versus effect size

(log-odds ratio) for variant-AF associations. Two gray dashed lines indicate MAFs of 0.001 and 0.01. The loci with an effect of odds ratio > 1.3 were labeled with the gene name. **c** Distribution of loci identified across GWAS of different ancestries. **d** Venn diagram of shared and unique loci across ancestries. Two loci near *PITX2* and *ZFHX3* were identified as shared across European (EUR), East Asian (EAS), African (AFR), and Admixed American (AMR) populations. Source data are provided as a Source Data file.

muscle development and heart contraction in AF onset while also uncovering additional pathways, including cellular response to transforming growth factor-beta (TGF-β), artery morphogenesis, regulation of cell communication via electrical coupling, and actin filament-based movement (Supplementary Data 6).

## Population-specific genetic correlations uncovered circulatory comorbidities

After Bonferroni correction, AF was significantly associated with 95 of 128 circulatory endpoints in Europeans (Supplementary Data 7) and 18 of 95 in Africans (Supplementary Data 8). Among the traits assessed for heterogeneity in genetic correlation with AF between European and African populations, several phenotypes demonstrated substantial population-specific differences. We identified conditions such as first-degree atrioventricular block, abdominal aortic aneurysm, varicose vein of lower extremity, deep vein thrombosis, tachycardia, transient cerebral ischemia, and abnormal heart sounds as having significantly heterogeneous genetic correlations with AF across ancestries (Fig. 4).

## Mendelian randomization revealed modifiable risk factors

Among the 37 modifiable risk factors, genetically predicted body mass index (BMI), waist-to-hip ratio, visceral adiposity, childhood BMI, apolipoprotein A-I levels, apolipoprotein B levels, low-density lipoprotein (LDL) cholesterol levels, type 2 diabetes, systolic and diastolic blood pressure, thyroid-stimulating hormone levels, smoking initiation, lifetime smoking index, alcohol consumption, leisure screen

time, and insomnia were significantly associated with AF risk after Bonferroni correction (Fig. 5). The scatter plots of the effect of SNPs on these traits and that on AF are shown in Supplementary Figs. 2–17. These associations remained robust in sensitivity analyses (Supplementary Data 9).

## Bidirectional protein-wide Mendelian randomization identified causal proteins

After pooling protein quantitative trait loci (pQTL) from deCODE and UKB-PPP, the forward Mendelian randomization (MR) analysis (the effect of genetically predicted protein levels on AF) included 2847 unique proteins with *cis* genetic variants as the instrumental variables. After filtering the association with $P < 0.05$ after Bonferroni correction, $P$ for heterogeneity in dependent instruments (HEIDI) test > 0.05, we identified genetically predicted levels of 95 circulating proteins were associated with AF risk (Fig. 6a and Supplementary Data 10). Among these, 21 and 16 protein-AF associations were identified as strong colocalization evidence with PPH4 > 0.8, respectively, using traditional colocalization (Fig. 6b and Supplementary Data 11) and Sum of Single Effects (SuSiE) colocalization (Fig. 6c and Supplementary Data 11) methods. In total, 28 proteins were deemed with potential causal associations with AF, with one standard deviation increment conferring an odds ratio of AF from 0.61 (95% CI 0.49–0.75) for ING1 to 1.68 (95% CI 1.35–2.09) for ATXN2L. Among these 28 proteins, 18 proteins had cis instruments available in the Fenland study, and 17 associations were replicated with $P$-value < 0.05 albeit the direction of

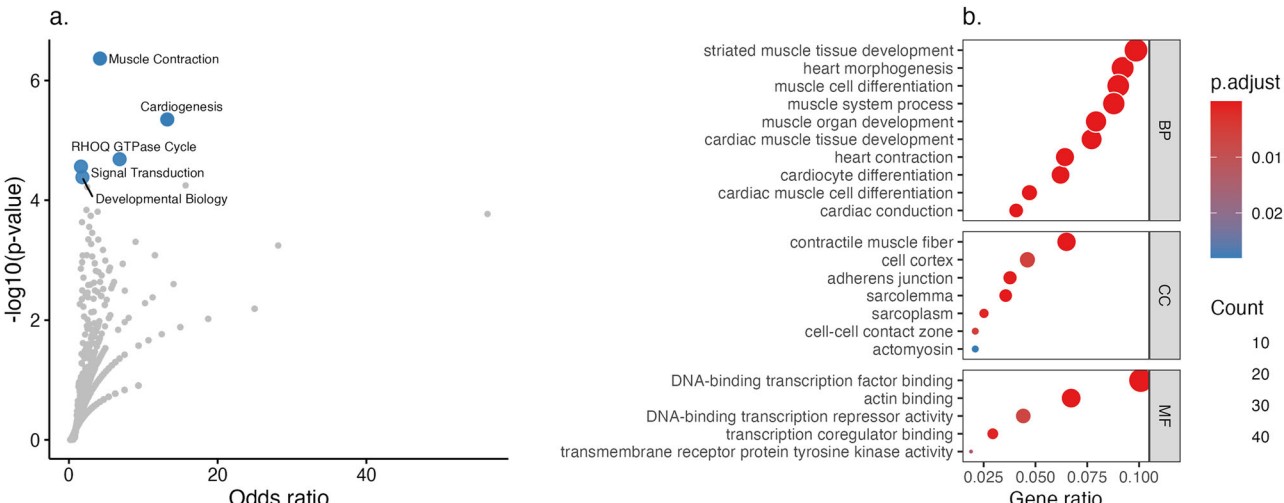

**Fig. 3 | Pathways enriched based on AF-associated genes. a** Pathway enrichment in the Reactome database. The *x*-axis represents the effect size of the pathway's influence on AF, while the *y*-axis shows the -log10(*P*) values, indicating statistical significance. Each dot corresponds to a pathway, with blue dots representing pathways that are significant after Bonferroni correction. **b** Pathway enrichment in the Gene Ontology (GO) database. The analysis includes pathways categorized under biological processes (BP), molecular functions (MF), and cellular components (CC). The *x*-axis represents the ratio of AF-associated genes to the total number of genes in each pathway, while the y-axis lists the pathways. Each dot represents a pathway, where the color reflects the Bonferroni-adjusted *p*-value, and the size indicates the count of AF-associated genes in each pathway. For clarity, the figure only highlights the top 10 out of 50 BP pathways due to space constraints. Full results, including all pathways, are provided in Supplementary Data 5 and Supplementary Data 6. The statistical test was two-sided, and the Bonferroni-corrected significance level was applied. Source data are provided as a Source Data file.

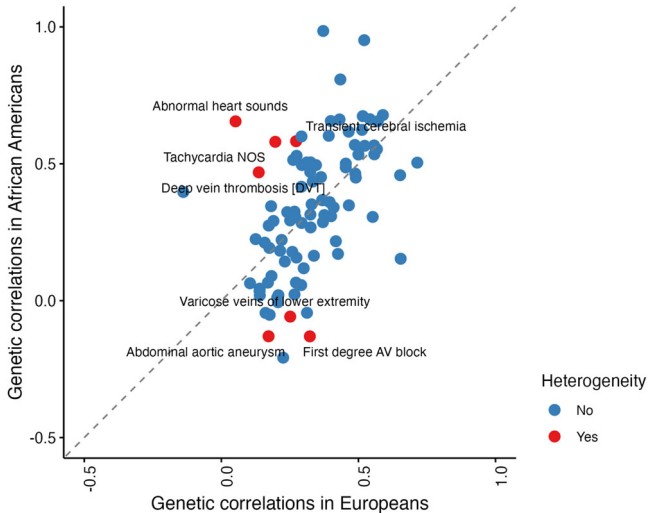

**Fig. 4 | Heterogeneity between Europeans and Africans regarding genetic correlations between atrial fibrillation and other circulatory endpoints.** The analysis was conducted using data from the Million Veteran Program (MVP). The analysis involved 94 correlations both in Europeans and Africans, and heterogeneity was defined by $I^2 > 75\%$ and *P*-value for Cochran's Q < 0.05. The statistical test was two-sided. Detailed information on these genetic correlations is available Supplementary Data 7 and 8. Source data are provided as a Source Data file.

the association was reverse for ICAM1, CCN3 (also known as NOV), and QSOX2 (Supplementary Data 12).

Seven protein targets have corresponding drugs in clinical trials or approved for other indications; however, none have been explicitly approved for treating AF (Supplementary Data 13). Nonetheless, certain targets, such as ICAM1, ANGPT1, and MAPK3, may hold therapeutic potential due to their roles in cardiovascular and inflammatory pathways, which are implicated in AF pathophysiology.

In the reverse MR analysis, genetic liability to AF was associated with levels of 16 unique proteins in deCODE or UKB-PPP (Supplementary Data 14 and 15) after Bonferroni correction. In particular, genetic liability to AF was associated with reduced levels of N-terminal pro-brain natriuretic peptide (NT-proBNP). The association for natriuretic peptide B was conflicting between deCODE and UKB-PPP.

**Polygenic risk and protein score enhanced disease prediction**
To evaluate the performance of the polygenic risk score in an independent dataset, we tested it in the Penn Medicine BioBank (PMBB), which is not used for PGS derivation. The polygenic risk score (PGS) derived from this cross-population GWAS meta-analysis demonstrated a dose-response association with AF prevalence in 4401 individuals with AF and 32,760 individuals without AF from the PMBB (Fig. 7a–c). Each standard deviation (SD) increase in PGS was associated with an odds ratio (OR) of 1.82 (95% CI: 1.79–1.85) for AF. Compared to individuals in the first decile of the PGS, those in the tenth decile had a sixfold increased risk of AF (OR = 6.38, 95% CI: 5.30–7.75) (Fig. 7b). Our PGS showed superior predictive performance compared to PGS002814 from the Miyazawa et al. study, with an area under a receiver operating characteristic (AUC) of 0.780 (95% CI: 0.778–0.783) and a Brier score of 0.092 (95% CI: 0.091–0.093), outperforming PGS002814 (AUC = 0.767, 95% CI: 0.764–0.769; Brier score = 0.094, 95% CI: 0.093–0.095) (Fig. 7c). The DeLong test showed that the AUC of the PGS derived from our GWAS meta-analysis was significantly higher than that of the Miyazawa PGS (*P* < 2.2e-16).

In a cohort of 3441 individuals with incident AF and 47,437 without, with available proteomic and genetic profiles, we constructed a protein score (ProS) using the LASSO method and a PGS to assess their predictive value for AF risk. The ProS included 87 proteins listed in **Supplementary Methods**. The ProS exhibited a positive association with AF incidence (Supplementary Data 16) and demonstrated strong predictive performance, achieving an AUC of 0.792 and a Brier score of 0.119 in the testing set (Fig. 7d). Similarly, the PGS also showed a robust association with incident AF (Supplementary Data 17). Adding the ProS to the PGS significantly enhanced the performance of AF risk prediction. The combined model incorporating PGS and ProS achieved an

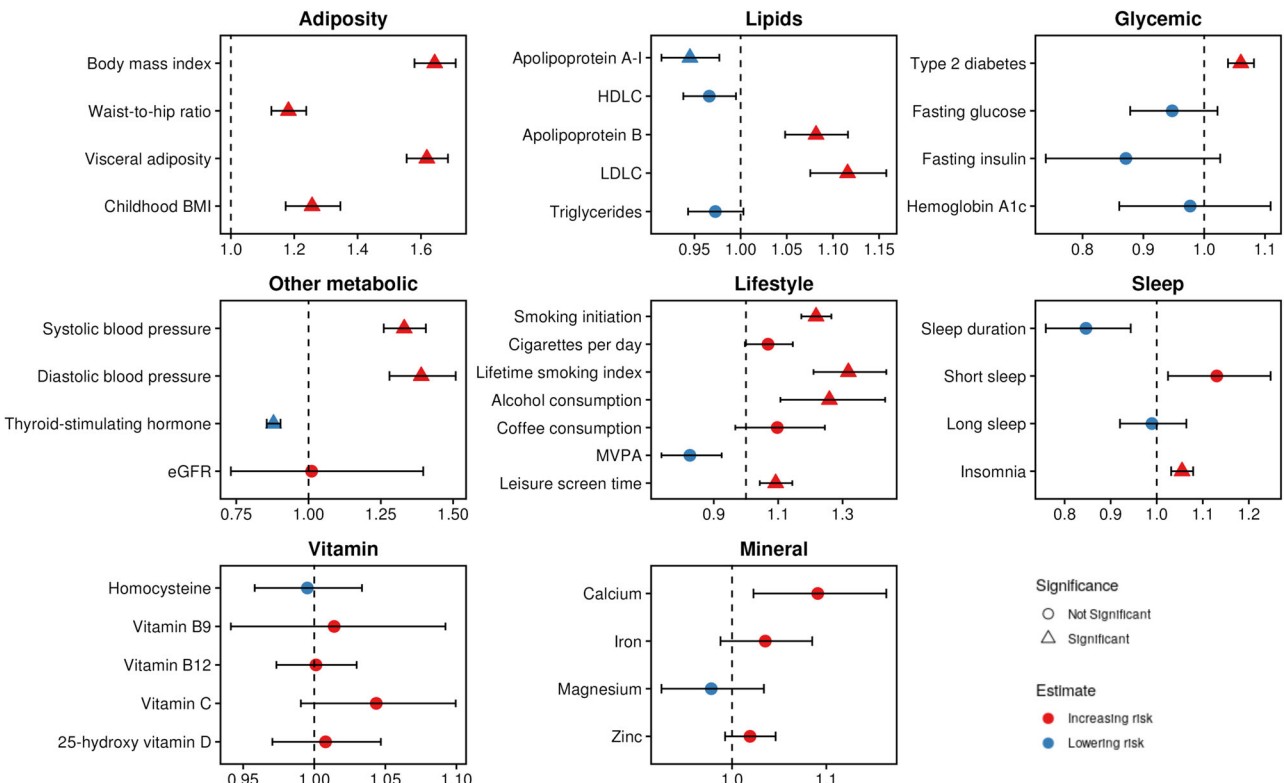

**Fig. 5 | Genetically predicted associations between 37 modifiable traits and atrial fibrillation (AF).** The estimates and p-values were derived using the inverse variance weighted (IVW) method with a fixed-effects model for traits with ≤ 4 genetic instruments. For traits with > 4 genetic instruments, the results were obtained from MR-PRESSO, accounting for potential pleiotropic effects by removing outlier SNPs where applicable. Detailed results are presented in Supplementary Data 9. Supplementary Data 18 lists the number of instrumental variables, the sample sizes of the source studies, and the units for each trait. The x-axis represents the odds ratio (OR) of AF per unit increase in the genetically predicted trait. Triangles indicate associations with $P < 0.05$ after Bonferroni correction, while red and blue dots represent positive and inverse associations, respectively. Data are presented as ORs +/− 95% confidence intervals. The statistical test was two-sided, and the Bonferroni-corrected significance level was applied. Source data are provided as a Source Data file.

AUC of 0.823 and a Brier score of 0.059 (Fig. 7d). The combined score incorporating both the PGS and ProS demonstrated superior predictive performance compared to either PGS alone ($P = 1.34 \times 10^{-21}$) or ProS alone ($P = 0.009$).

## Discussion

In this large-scale cross-population GWAS meta-analysis of AF, comprising 252,438 cases and 1,959,739 controls, we identified numerous previously unreported genetic loci, refined the genetic architecture of AF, and emphasized the importance of population-inclusive research in uncovering both shared and population-specific risk variants. Notably, our population-specific analysis revealed significant disparities in genetic risk loci, with a majority identified in Europeans and relatively few in non-Europeans. This imbalance likely reflects differences in sample sizes across ancestries, underscoring the urgent need to increase representation of underrepresented populations in future genetic studies to ensure equitable and comprehensive genetic discovery[9].

While most risk alleles had small-to-moderate effect sizes, we identified six lead SNPs with larger effect sizes in loci prioritized by *SORCS3, POLD1, AGBL4, AC126283.1, PITX2,* and *FAM241A* genes, suggesting stronger genetic contributions at these loci. *PITX2* has a well-documented role in AF through mechanisms involving electrical and structural remodeling, as well as calcium handling[10–12]. *AGBL4* has been revealed to be associated with AF in previous GWASs[7,13]. However, the involvement of *SORCS3, POLD1, AC126283.1,* and *FAM241A* in AF remains to be clarified through future studies.

*PITX2* and *ZFHX3* are well-established AF-associated genes; our findings reaffirm their consistent association across four population groups[7,8], further supporting their pivotal role in AF susceptibility. Regarding mechanisms, a knockout mice study revealed that *ZFHX3* loss in mice leads to atrial dysfunction, arrhythmogenic remodeling, and increased AF susceptibility[14]. However, no drugs targeting the two gens have been proved or developed, thus whether these two targets can be used for therapeutic development needs to be investigated.

We employed a comprehensive gene prioritization strategy, identifying putative causal genes for 504 loci, providing functional insights into AF pathogenesis. This approach enhanced pathway enrichment analyses, reaffirming muscle contraction and cardiac development[7] as core AF mechanisms while uncovering additional pathways, including TGF-β signaling, vascular remodeling[15], electrical coupling, and cytoskeletal regulation[16]. These findings highlight potential therapeutic opportunities, such as targeting TGF-β-mediated fibrosis or refining anti-arrhythmic strategies through ion channel modulation[17], paving the way for potential interventions in AF prevention and treatment.

We observed significant heterogeneity in the genetic correlation between AF and several circulatory phenotypes across European and African populations. While these findings suggest the possibility of population-specific differences in the shared genetic architecture between AF and its comorbidities, we acknowledge that these results are exploratory and require validation in independent cohorts. Due to the limited number of prior genetic studies addressing population-specific correlations for these traits, we refrain from drawing strong conclusions about the direction or clinical implications of individual trait differences. Instead, our findings underscore the broader need for population-informed genetic analyses and increased representation of diverse populations. Facilitating this type of research may improve the

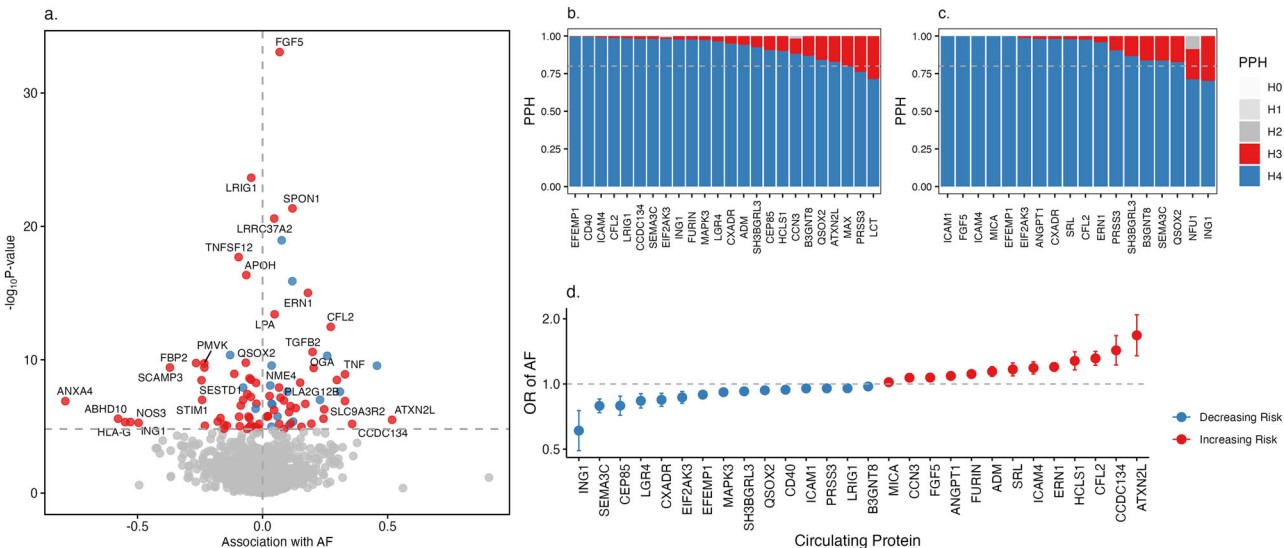

**Fig. 6 | Genetically predicted levels of 2847 proteins associated with atrial fibrillation (AF).** We analyzed 2847 unique proteins with *cis*-instrumental variables derived from the deCODE and UKB-PPP datasets. For proteins present in both datasets, data from UKB-PPP were prioritized due to its larger sample size. All associations were scaled to a one standard deviation increase in genetically predicted protein levels. **a** volcano plot of protein-AF associations using SMR analysis. The x-axis represents the effect size of protein-AF associations, while the y-axis shows the -log10($P$) values. The statistical test was two-sided, and the Bonferroni-corrected significance level was applied. Associations with $P < 0.05$ after Bonferroni correction and HEIDI test $P > 0.05$ are labeled. Red and blue dots indicate positive

and inverse associations, respectively. **b** traditional colocalization analysis results. Only protein-AF associations with PPH4 > 0.7 are displayed due to space constraints. The gray line indicates PPH = 0.8, a commonly used threshold for strong colocalization evidence. **c** SuSiE colocalization analysis results. Similar to panel b, only protein-AF associations with PPH4 > 0.7 are shown. The gray line indicates PPH = 0.8. **d** forest plot of associations meeting the criteria of Bonferroni-corrected $P < 0.05$, HEIDI $P > 0.05$, and colocalization PPH4 > 0.8. Data are presented as ORs +/− 95% confidence intervals. The statistical test was two-sided, and the Bonferroni-corrected significance level was applied. Source data are provided as a Source Data file.

accuracy of risk stratification, inform targeted screening strategies, and reduce disparities in cardiovascular outcomes across diverse patient populations.

Our MR analyses identified obesity[18], type 2 diabetes[19], hypertension[20], high TSH levels[21], smoking[22], and insomnia[23] as causal risk factors for AF, consistent with previous studies. However, for dyslipidemia[24], alcohol consumption[22], and sedentary behavior[25]—traits with conflicting evidence in prior research—our well-powered MR analysis leveraging a larger sample size strengthened their associations with AF. Mechanistically, obesity, lipid imbalances, and hypertension may drive atrial remodeling and inflammation, while smoking, alcohol consumption, and insomnia could exacerbate autonomic dysfunction and electrical instability, increasing AF susceptibility. Clinically, these findings emphasize the need for targeted AF prevention strategies, including weight management, lipid-lowering therapies, blood pressure control, and behavioral interventions to reduce sedentary behavior. Addressing these modifiable risk factors through lifestyle changes and medical interventions could play a crucial role in reducing AF incidence and its associated complications.

Our study identified 28 circulating proteins with potential causal roles in AF, some of which have been previously associated with the condition[26,27]. Among these, our MR associations for ICAM1[28] and CD40[29] were directionally opposite to prior observational studies, likely reflecting compensatory or feedback mechanisms[30,31]. The positive association for FURIN aligns with its role in pro-fibrotic and inflammatory pathways[32], while ADM's association supports its involvement in vascular regulation[33], both of which may contribute to AF onset. Although none of these proteins have been established as direct therapeutic targets for AF, our findings provide valuable insights into AF pathophysiology and highlight promising candidates for further investigation[30,34]. Nonetheless, we observed that a subset of associations could not be replicated in the independent dataset. While such discrepancies may arise from differences in genetic regulation across populations, platform-specific variation in protein quantification, or

measurement error in replication analyses, they do not necessarily invalidate the MR results. However, they do warrant caution in interpreting these findings. Importantly, the associations identified in our study are based on protein levels measured in circulation and may not fully capture tissue-specific effects relevant to AF pathogenesis. Further validation in independent cohorts and functional characterization of these proteins in cardiac-relevant tissues and models will be essential to confirm their causal roles and assess their translational potential.

MR revealed that genetic liability to AF is paradoxically associated with lower circulating NT-proBNP levels, in direct contrast to case-control studies reporting elevated NT-proBNP among AF patients[35,36]. This discordance implies that the NT-proBNP elevations seen in AF may largely reflect secondary hemodynamic stress and atrial stretch rather than a primary effect of AF itself. Moreover, we observed inconsistent associations between genetic liability to AF and NPPB—the prohormone precursor to NT-proBNP—across two independent proteomic datasets, underscoring additional complexity. Although longitudinal cohorts have linked higher baseline NT-proBNP to subsequent AF[37], our SMR analyses did not support a causal influence of genetically proxied NT-proBNP or NPPB on AF risk. Together, these data argue against a simple, unidirectional causal relationship between AF and NT-proBNP, and highlight the need for detailed longitudinal and mechanistic studies to untangle cause from consequence in the AF–NT-proBNP axis.

Our study highlights the strong predictive value of a PGS derived from a cross-population GWAS, demonstrating superior performance compared to previous PGSs[8]. The enhanced predictive accuracy has significant implications for risk differentiation at the population level. In addition, we developed a ProS and found that combining the ProS with the PGS significantly improved risk prediction, aligning with findings from prior studies. A UK Biobank-based study demonstrated improved disease prediction when integrating a protein score with a clinical score[38], while another UK Biobank study found a significant improvement when combining a protein score with a PGS[39]. Even

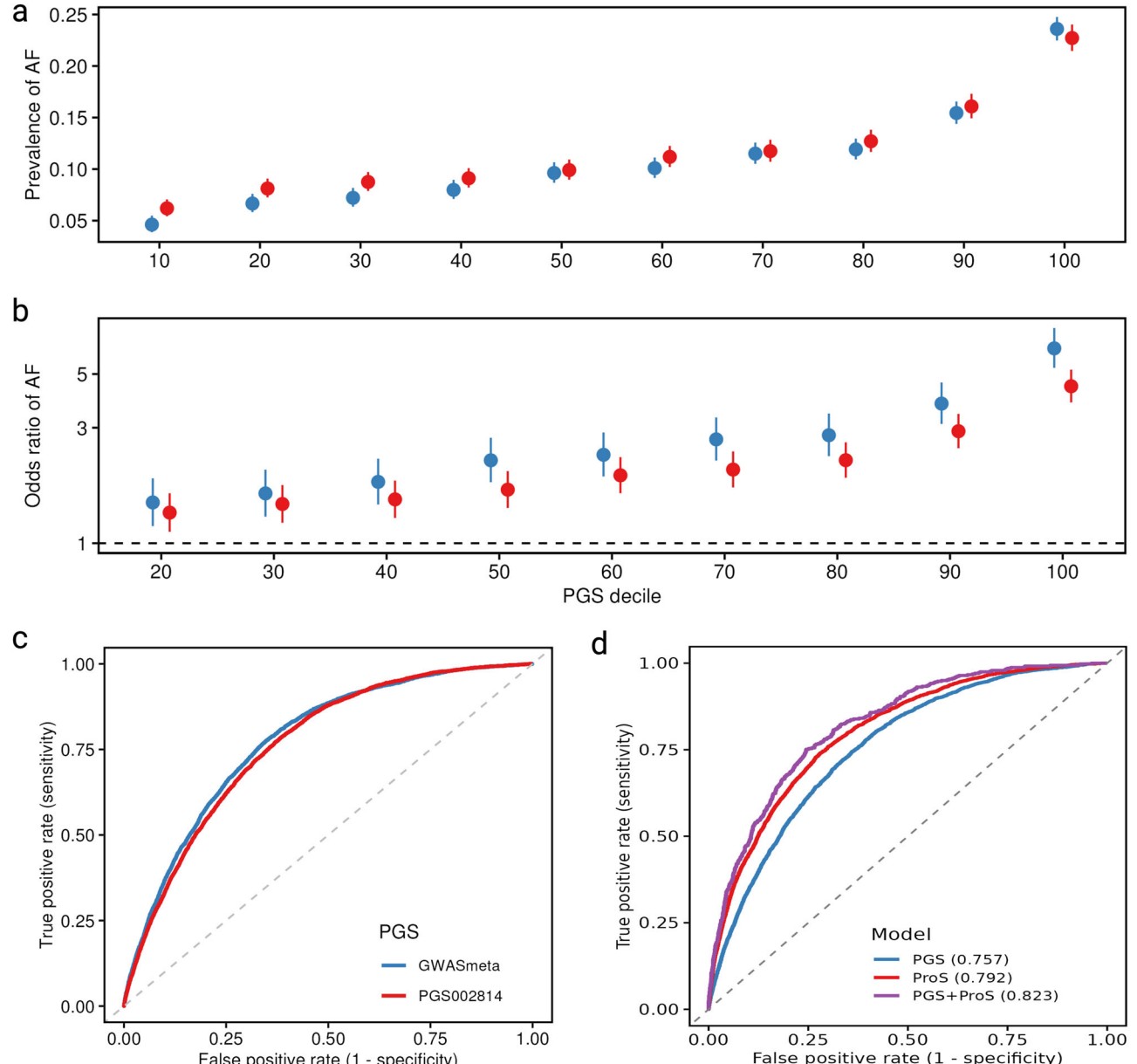

**Fig. 7 | Polygenic risk score (PGS) and protein score (ProS) for atrial fibrillation (AF) risk prediction.** The analysis for panels (**a**, **b**, and **c**) was based on the Penn Medicine Biobank (PMBB, 4401 individuals with prevalent AF and 32,760 individuals without) and the analysis for panel d was based on the UK Biobank (3441 individuals with incident AF and 47,437 without). Panels (**a** and **b**) plots show the prevalence and odds ratio of AF across deciles of our PGS vs. the PGS002814 from the Miyazawa et al. study, respectively. Data in panels (**a** and **b**) are presented as mean values +/− SD and ORs +/− 95% confidence intervals, respectively. Panel c plot compares the prediction ability between two PGS (AUC for our PGS = 0.780 and AUC for PGS002814 = 0.767). Panel (**d**) plot compares the prediction ability between PGS, ProS, and their combination. AUC, area under its receiver operating characteristic curve. Source data are provided as a Source Data file.

though different protein selection methods were used between previous studies and the current study, the findings remained consistent. Collectively, these results underscore the value of multi-omic approaches in refining AF risk assessment. Future research should focus on validating these models in diverse populations and evaluating their potential clinical applications to further enhance personalized AF prevention and management strategies.

This study has several limitations. First, although we included data from non-European populations, the statistical power for these ancestries may be limited due to smaller sample sizes, potentially affecting the identification of population-specific associations. Second, despite employing multiple prioritization strategies, some degree of gene misassignment is likely inevitable due to the limitations of current functional annotation resources. While many genes at these loci were prioritized based on proximity to the lead variant, we have explicitly noted when proximity was the sole criterion and, where possible, incorporated supporting evidence from eQTL colocalization and fine mapping to strengthen biological plausibility. Third, although we applied a MAC < 50 threshold to exclude rare variants, a small number of variants with minor allele frequency (MAF) < 1% remained in the analysis (5 out of 493 in European GWAS and 7 out of 525 in cross-population GWAS). However, nearly half of these variants were replicated in our independent replication dataset or have been previously reported in association with AF in other studies. Given their limited number and supporting evidence, we believe that the inclusion of these variants does not materially affect the comparability of our results with earlier GWAS. Fourth, while there were some sample overlaps in the MR analysis, the potential bias is likely minimal due to the small proportion of overlapping samples and the

strong validity of the genetic instrumental variables used. Fifth, the inclusion of coding variants may alter epitope binding in aptamer-based proteomic analyses for certain proteins, potentially introducing measurement bias that could affect the accuracy of MR results[40]. Lastly, all analyses were conducted using in silico approaches, emphasizing the need for further validation through functional studies and experimental research to confirm the biological relevance of the identified associations.

In summary, this cross-population GWAS meta-analysis identified 525 genetic loci for AF, refining its genetic architecture and biological pathways. Mendelian randomization revealed causal risk factors and circulating proteins, offering insights for prevention and therapeutic development. The cross-population-derived PGS, combined with a protein score, significantly improved risk prediction. This study integrates genetic discovery, causal inference, and multi-omic data, advancing AF risk stratification, prevention, and potential therapeutic strategies.

## Methods

### Ethics
The study complied with all relevant regulations governing the use of human participants and was conducted in accordance with the principles of the Declaration of Helsinki. Participants in the FinnGen study provided informed consent for biobank research, with the study protocol (No. HUS/990/2017) approved by the Coordinating Ethics Committee of the Hospital District of Helsinki and Uusimaa (HUS). The UK Biobank received ethical approval from the North West Multi-center Research Ethics Committee (approval number: 11/NW/0382), with all participants giving informed consent. The Million Veteran Program (MVP) was approved by the VA Central Institutional Review Board (IRB), and participants provided informed consent. The Penn Medicine Biobank (PMBB) was approved by the University of Pennsylvania Institutional Review Board, and all participants gave informed consent. The Swedish Ethical Review Authority granted ethical approval for SIMPLER and the current protocol (no. 2019-03986), and all participants gave informed consent. Each study adheres to rigorous ethical guidelines to ensure the protection of participants and the integrity of the research.

### Study design and participants
Figure 1 summarizes the study design. We first performed a GWAS meta-analysis across eight studies as the discovery analysis in European populations. This was followed by a replication analysis using data from the UK Biobank, resulting in a European GWAS meta-analysis that included a total of 228,926 AF cases and 1,611,415 controls. Next, we extended the analysis to include data from East Asians, South Asians, Africans, and Admixed Americans, enabling a cross-population meta-analysis comprising 252,438 AF cases and 1,959,739 controls. Detailed descriptions of included studies are shown in **Supplementary Methods** and Supplementary Data 1. Using this large-scale AF GWAS, we conducted comprehensive downstream analyses to prioritize related genes, explore potential etiologies, assess genetic correlations, identify risk factors, and evaluate risk prediction models.

### Cross-population GWAS meta-analysis
Eight studies (the Nord-Trøndelag Health Study [HUNT], deCODE, DiscoverEHR, Michigan Genomics Initiative [MGI], AFGen consortium[4], FinnGen R12, Swedish Infrastructure for Medical Population-Based Life-Course and Environmental Research [SIMPLER, https://www.simpler4health.se/], and Million Veteran Program[MVP]) contributed to the discovery analysis for the European GWAS, comprising 192,851 atrial fibrillation (AF) cases and 1,239,541 controls[6,41,42]. We performed GWAS association testing using individual-level genotype and phenotype data from participants in the SIMPLER cohort. By incorporating replication data from the UK Biobank (36,075 cases and 371,874 controls), the total sample size for the European GWAS reached 228,926 cases and 1,611,415 controls. To expand the analysis, we included data

from four additional ancestries represented in Biobank Japan[43], Genes & Health[44], and MVP[41], culminating in a cross-population meta-analysis with 252,438 AF cases and 1,959,739 controls. Detailed descriptions of the study populations, genotyping procedures, and quality control protocols are provided in the **Supplementary Methods**, while AF definitions and sample sizes for each included study are summarized in Supplementary Data 1.

Each dataset underwent rigorous quality control, including initial preprocessing, genotype imputation, post-imputation filtering, and association testing, with adjustments for age (or birth year), sex, and principal components as covariates. Post-GWAS quality control was performed using GWASinspector[45], and SNPs with minor allele counts < 50 were excluded. Meta-analyses were conducted using METAL[46], employing the fixed-effect inverse-variance-weighted method. After meta-analysis, variants that were present in only one cohort were excluded from downstream analysis.

We applied LDSC to evaluate the contributions of population stratification and polygenicity to GWAS test statistic inflation[47]. Although the genomic inflation factor (λGC) was 2.04, the LDSC intercept (1.34) and ratio (15%) indicated that most of the inflation could be attributed to a true polygenic signal rather than confounding biases. Genome-wide significant SNPs were grouped into loci if they were within 1 Mb of each other[8]. Loci were defined by (1) identifying genome-wide significant variants ($P < 5 \times 10^{-8}$) from association results, (2) extending the region by 500 kb on either side of these variants, and (3) merging overlapping regions. Genetic loci in the European analysis were defined based on a GWAS meta-analysis that combined both the discovery and replication datasets. This integrated approach maximized statistical power, enabling the identification of several loci that reached genome-wide significance only after the datasets were meta-analyzed. Loci were annotated as unreported if loci had no overlapping coordinates with previously reported genome-wide significant variants ($P < 5 \times 10^{-8}$) associated with AF based on a comprehensive evaluation. This included PheWAS lookups using the Open Targets platform (https://genetics.opentargets.org/, integrating data from the GWAS Catalog, UK Biobank, and FinnGen), as well as cross-referencing with prior AF GWAS reports, including those by Thorolfsdottir et al. (2017)[48], Nielsen et al. (2018)[6], Roselli et al. (2018, 2025)[7,49], Miyazawa et al. (2023)[8], Verma et al. (2024)[41], Choi et al. (2025)[50], and other relevant studies.

### Gene prioritization
We applied six complementary gene prioritization approaches to identify the most confident locus-gene pairs: (1) nearest gene annotation, (2) MAGMA-based gene prioritization[51], (3) Polygenic Priority Score (PoPS)[52], (4) eQTL colocalization, (5) CARMA (Credible-variant Analysis for Regional Meta-Analysis)-based functional gene prioritization[53], and (6) transcriptome-wide association study (TWAS)[54]. For each genomic locus, the prioritized gene was determined by selecting the gene with the highest count of selections across these six methods. In cases where multiple genes had the same count, prioritization was refined by first considering genes encoding variants within CARMA-identified credible sets, followed by the nearest gene[55]. Below is a detailed description of each approach:

### Nearest gene annotation
The gene closest to the lead SNP in each locus was identified based on its physical distance to the gene body. This analysis was performed using the get_nearest_gene() function from the gwasRtools R package (https://github.com/lcpilling/gwasRtools).

### MAGMA-Based gene prioritization
We utilized MAGMA to annotate genes within genomic loci using the 1000 Genomes Project as the reference panel[51]. SNPs were mapped to genes based on their physical positions, including the gene body and

flanking regions (± 10 kb). Gene-level *p*-values were then calculated by aggregating SNP association statistics while accounting for linkage disequilibrium (LD) structure. The gene with the smallest *p*-value within each locus was selected as the prioritized gene.

## PoPS

PoPS, a similarity-based gene prioritization tool, integrates publicly available datasets, such as RNA sequencing data, curated pathway annotations, and predicted protein-protein interaction networks[52]. Based on the premise that causal genes share similar functional characteristics, PoPS calculates gene-level association statistics using GWAS summary statistics and MAGMA-based gene annotations. It then selects relevant features from precomputed statistics and assigns a score to each gene, reflecting its likelihood of being causal. For each genome-wide significant locus, genes within 1 Mb of the index variant (in both directions) were ranked by their PoPS scores, with the highest-ranked gene prioritized.

## eQTL Colocalization

Colocalization analysis was conducted using the coloc R package, which applies an approximate Bayes factor framework to assess whether two traits share a causal genetic signal[56]. Using the coloc.abf() function (https://github.com/chr1swallace/coloc), we calculated posterior probabilities for five hypotheses: (H0) no association with either trait; (H1/H2) association with only one trait; (H3) association with both traits but different causal variants; and (H4) association with both traits with the same causal variant. A high posterior probability for H4 (PP4 > 0.8) was considered evidence of colocalization[57]. For this analysis, we used eQTL data from eQTLGen Phase I[58] and the Genotype-Tissue Expression (GTEx) Project v8[59] for heart atrial appendage and heart left ventricle tissues. Variants within 500 kb of each GWAS index variant were extracted to perform colocalization analysis.

## CARMA-Based functional gene prioritization

We applied CARMA, a Bayesian fine-mapping approach[53], to identify credible sets of variants within each genomic locus. CARMA accounts for LD structure and aggregates association signals across studies or populations to identify variants most likely to be causal. For each locus, CARMA generated a credible set with a high posterior probability (e.g., 95%) of containing the causal variant(s). Functional annotation of these variants was performed using Open Targets (https://www.opentargets.org/), which provides information on coding, regulatory, and splicing effects[60]. If a causal variant was located within or directly affected a gene's function, that gene was assigned to the locus.

## TWAS

We performed TWAS using MetaXcan[54] to estimate the relationship between genetically predicted gene expression and AF. MetaXcan integrates GWAS summary statistics with precomputed gene expression prediction models to identify genes associated with the phenotype. For this analysis, we used expression prediction models for the heart atrial appendage, artery tibial, and heart left ventricle, leveraging LD reference data from GTEx v8[59] and cross-population AF-GWAS summary statistics. For the TWAS, the target tissues were selected based on results from MAGMA tissue enrichment analysis and stratified-LDSC[61], both of which were conducted using gene expression data from GTEx v8. MAGMA tissue enrichment analysis identifies tissues where genes associated with the trait of interest are significantly enriched by testing the relationship between GWAS association signals and tissue-specific gene expression profiles. S-LDSC further refines this by partitioning heritability across genomic regions annotated with tissue-specific gene expression and estimating the contribution of each tissue to the trait heritability. Using these complementary approaches, tissues such as the heart atrial

appendage, heart left ventricle, and artery tibial were identified as relevant for atrial fibrillation (Supplementary Fig. 18). These selected tissues were then used to predefine the expression prediction models for the TWAS. Bonferroni correction was applied to account for multiple testing, and the gene with the lowest *p*-value within each locus was prioritized.

## Pathway enrichment

Pathway enrichment analysis was performed to identify biological pathways and functional categories associated with the prioritized genes. Reactome[62] enrichment was conducted using Enrichr (https://maayanlab.cloud/Enrichr/), enabling the exploration of curated pathways[63]. Gene Ontology (GO) enrichment analysis[64], which provided insights into biological processes, molecular functions, and cellular components, was carried out using the enrichGO function from the clusterProfiler Bioconductor R package (https://bioconductor.org/packages/release/bioc/html/clusterProfiler.html). To minimize false-positive findings, Bonferroni correction was applied to account for multiple testing, with the significance threshold set at $P < 0.05$/number of tests performed.

## Population-specific genetic correlations with circulatory endpoints

Using LDSC, we calculated the genetic correlations of AF with 130 and 97 circulatory endpoints defined by phecodes, separately for Europeans and Africans in the MVP cohort. The MVP GWAS included up to 449,042 European participants and 121,177 African participants[41]. Genetic correlations with $rg > 1.25$ or $< -1.25$ were removed due to poor inheritability ($h^2$ estimates was very close to zero). To account for multiple testing and reduce the likelihood of false-positive results, the Bonferroni correction was applied. To objectively compare genetic correlations between populations, we applied Cochran's Q test to assess heterogeneity in the correlation estimates between European and African populations. Traits were considered to exhibit population-specific differences if they showed evidence of substantial heterogeneity, defined as an $I^2$ statistic greater than 75% and a Cochran's Q test $P$–value less than 0.05.

## Mendelian randomization analysis for modifiable risk factors

MR is an analytical approach that strengthens causal inference by leveraging genetic variants (IVs) as instrumental variables to estimate the causal effect of an exposure on an outcome. A comprehensive description of the MR design is provided in the **Supplementary Methods**. Using GWAS meta-analysis data, we conducted MR to evaluate the associations between 37 modifiable risk factors and AF risk. These modifiable factors span multiple categories, including adiposity, blood lipids, type 2 diabetes and glycemic traits, other metabolic traits (e.g., blood pressure, thyroid function, and kidney function), lifestyle factors (e.g., smoking, alcohol and coffee consumption, and physical activity), sleep behaviors, and dietary factors (e.g., circulating levels of vitamins and minerals). The selection of these factors was guided by a recent comprehensive review of AF risk factors[65]. Detailed information on the GWAS data sources for these traits is summarized in Supplementary Data 18.

Genetic variants associated with the exposures were selected at a genome-wide significance threshold of $P < 5 \times 10^{-8}$. To ensure independence among instrumental variables, SNPs were pruned at $R^2 < 0.01$, minimizing the effects of collinearity due to LD. The strength of the instrumental variables was assessed using F-statistics[66], with all variants meeting the threshold of $F > 10$. Data harmonization was performed to align effect and non-effect alleles consistently between the exposures and outcomes. Detailed information on the used genetic instruments is presented in Supplementary Data 19.

For exposures with fewer than five genetic instruments, the inverse variance weighted (IVW) method with a fixed-effects model

was used. For exposures with five or more genetic instruments, we employed MR-PRESSO as the primary analysis method, as it accounts for pleiotropic effects by identifying and removing outlier SNPs[67]. In the absence of outlier SNPs, MR-PRESSO provides estimates equivalent to the IVW method. Sensitivity analyses included the IVW method with random effects, the weighted median method[68], and MR-Egger regression[69]. Heterogeneity among SNP-specific estimates was assessed using Cochran's Q test, while the MR-Egger intercept test was used to evaluate the presence of horizontal pleiotropy. The scatter plot was used to visualize potential pleiotropic SNPs. To minimize false-positive findings, we applied Bonferroni correction to account for multiple testing.

### MR and colocalization analyses for circulating proteins

For the MR analysis of circulating proteins, we utilized two large-scale pQTL (protein quantitative trait loci) datasets, deCODE[70] and UKB-PPP[71], for IV selection (Supplementary Fig. 19). After excluding overlapping proteins, a total of 2847 proteins with *cis*-SNPs were included in the analysis. For proteins present in both datasets, we prioritized data from UKB-PPP due to its larger sample size and the fact that it identified a greater number of *cis*-pQTLs using the Olink[72]. Importantly, the associations of overlapping proteins with AF showed strong consistency between the two datasets, supporting the robustness of the findings. To validate the results, we used the Fenland study as a replication dataset[73], focusing on proteins with available IVs in this study to replicate the observed associations.

We used the lead *cis*-SNP associated with plasma protein levels at $P < 5 \times 10^{-8}$ as the genetic IV. *Cis*-SNPs were defined as variants located within 250 kb of the encoding gene. Detailed information on selected genetic IVs is shown in Supplementary Data 20. The Summary-data-based Mendelian Randomization (SMR) method was employed to estimate the association between genetically predicted protein levels and AF risk[74]. SMR integrates GWAS and pQTL summary statistics to evaluate whether the genetic association with a phenotype (i.e., AF) is mediated through the genetically regulated protein levels. To evaluate potential pleiotropy, we performed HEIDI (Heterogeneity in Dependent Instruments) analysis[74]. HEIDI assesses whether the association between the protein and the phenotype is driven by the same causal variant or by independent variants in LD. The analysis uses 3–20 SNPs in the *cis* region of the encoding gene to test for heterogeneity. A HEIDI *p*-value > 0.05 suggests no evidence of pleiotropy and supports the hypothesis of a shared causal variant. To further rule out false-positive associations caused by LD, we conducted traditional[56] and SuSiE (Sum of Single Effects)[75] colocalization analyses, using all SNPs in the *cis* gene region as input. As described in detail in the eQTL colocalization section, strong evidence of shared causal variants between protein levels and AF was indicated by PP.H4 ≥ 0.8, a stringent but widely accepted threshold in colocalization studies. We applied Bonferroni correction to account for multiple testing. Associations were considered potentially causal if they met the following criteria: adjusted $P < 0.05$ for the SMR analysis, adjusted $P > 0.05$ for the HEIDI test (indicating no pleiotropy), and colocalization posterior probability PP.H4 ≥ 0.8. Bonferroni correction was used for multiple testing for SMR analysis.

The druggability of identified proteins was assessed using multiple drug databases, including DrugBank[76], DepMap[77], and OpenTargets[60]. Based on their therapeutic potential, proteins were classified into five categories: (1) approved drug targets, (2) in clinical trials, (3) preclinical candidates, (4) druggable, and (5) not currently listed as druggable targets.

To examine the effect of genetic liability to AF on blood proteins, we conducted a reverse MR analysis using 624 SNPs as instrument variables for AF ($P < 5 \times 10^{-8}$ and $r^2$ for linkage disequilibrium < 0.01)

and protein GWAS data from deCODE and UKB-PPP. Bonferroni correction was used for multiple testing.

### Joint performance of PGS and protein score (ProS)

**PGS analysis.** The weights of the polygenic scores (PGS) in the current study were generated using the "auto" setting of PRS-CSx[78], incorporating summary statistics from the meta-analysis and corresponding EUR, AFR, AMR, EAS, or SAS LD reference panels derived from 1000 Genomes Project Phase 3 samples. This approach eliminates the need for independent training data. The effective sample size was calculated as 4/((1/ncases) + (1/ncontrols)). For the score applied to the UK Biobank, weights were derived using data that excluded summary statistics from UK Biobank participants. As a reference, we used the PGS (PGS002814) from the Miyazawa et al. study[8], which was derived using the Pruning and Thresholding method ($r^2 = 0.5$ and $P = 5 \times 10^{-4}$; https://www.pgscatalog.org/score/PGS002814/). We used the DeLong test[79], implemented in the pROC R package, to statistically compare the AUROC of the polygenic scores and evaluate whether the difference in predictive performance was significant. We calculated PGS for 4401 individuals with AF and 32,760 individuals without from the Penn Medicine BioBank (PMBB)[80], an ongoing study that integrates genomic and electronic health record data to investigate the genetic and clinical determinants of various diseases. The population breakdown of PMBB participants is predominantly European ($\sim 70\%$), followed by African ($\sim 25\%$), with smaller proportions of South Asian, East Asian, Admixed American, and other populations (Supplementary Fig. 20). The study was approved by the University of Pennsylvania Institutional Review Board. To standardize the scores, we applied a principal component analysis-based method, normalizing both the mean and variance to the 1000 Genomes reference panel. The association between PGS and prevalent AF was assessed using a generalized linear regression model with a logit link, adjusting for age and sex as covariates. We evaluated the PGS effect size using odds ratios and assessed model performance by calculating the area under the receiver operating characteristic curve (AUROC) and Brier score. Using the 'tidymodels' R package (https://github.com/tidymodels), we performed V-fold cross-validation to validate model performance. The same approach was used to test PGS performance in the UK Biobank, including 3441 individuals with incident AF and 47,437 without incident AF with available proteomic profiles.

**Protein score analysis.** We derived a protein score (ProS) for AF using individual-level data from the UK Biobank, a large, ongoing population-based prospective cohort study with extensive proteomic and phenotypic data. To rule out proteins with reverse associations, we first conducted a prospective cohort analysis. Participants with baseline AF or those diagnosed with AF within the first two years of follow-up were excluded, leaving 50,878 participants with proteomic data. Proteins with a missing rate exceeding 30% were also excluded, resulting in a final dataset of 2920 proteins. After adjusting for age, sex, ethnicity, Townsend deprivation index, education, body mass index, smoking status, drinking status, and physical activity, 459 proteins were significantly associated with incident AF after Bonferroni correction (Supplementary Data 21). We then used the Least Absolute Shrinkage and Selection Operator (LASSO) method to construct the ProS[81]. We applied LASSO logistic regression to identify candidate proteins associated with AF, using five-fold cross-validation to determine the optimal penalty parameter (λ). A weighted protein score (ProS) was then constructed based on the proteins selected via LASSO. Specifically, a Cox regression model was used to estimate the log-hazard ratios for each protein and the baseline hazard function. The individual risk score for each participant was subsequently calculated using: *Risk Score* $= h_0(t) \times exp(\beta_1 X_1 + \beta_2 X_2 + \cdots + \beta_n X_n)$, where $X_n$ is the level of the n-th selected protein, and $\beta_n$ is the corresponding coefficient from the Cox model. Participants were randomly split into

training and validation cohorts in a 7:3 ratio using the R package caret (https://github.com/topepo/caret). The model demonstrating the best predictive performance in the training cohort were then validated in the remaining 30% of participants and ultimately combined into a final model for predicting the risk of AF onset.

**Joint performance.** The AUROC analysis was performed to assess the predictive performance of the selected key proteins for AF, both individually and in combination with the PGS in the UK Biobank. The DeLong test[79] was used to statistically compare the AUROC of these scores and their difference in predictive performance.

### Reporting summary
Further information on research design is available in the Nature Portfolio Reporting Summary linked to this article.

## Data availability
The UK Biobank data are available through an application (https://www.ukbiobank.ac.uk/). The summary-level data of FinnGen (https://storage.googleapis.com/finngen-public-data-r12/summary_stats/release/finngen_R12_I9_AF.gz) and MVP (https://ftp.ncbi.nlm.nih.gov/dbgap/studies/phs002453/analyses/GIA/) are publicly online. The GWAS data generated in this study have been deposited in the NHGRI-EBI GWAS catalog database under accession codes GCST90624411 (cross-population GWAS), GCST90624412 (European GWAS), GCST90624413 (European GWAS after excluding UK Biobank), and GCST90624414 (UK Biobank GWAS). Source data are provided in this paper.

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

## Acknowledgements

We want to acknowledge the participants and investigators of the UK Biobank study, SIMPLER, the FinnGen study, the Veterans Affairs Million Veteran Program, and the Penn Medicine Biobank. This research was conducted using the UK Biobank study under application Number 72723. SIMPLER receives funding through the Swedish Research Council under grant numbers 2017-00644, 2017-06100 and 2021-00160, and from Stiftelsen Olle Engkvist Byggmästare (SOEB). The computations were performed on resources provided by SNIC through Uppsala Multidisciplinary Center for Advanced Computational Science (UPPMAX) under Project simp2021005. This work was supported by research grants from the Swedish Heart Lung Foundation (Hjärt-Lungfonden no. 20210351 and no. 20240457; to Susanna C. Larsson). Shuai Yuan was supported by an Award from the American Heart Association and the VIVA Foundation (https://doi.org/10.58275/AHA.24POST1189614.pc.gr.190880; Award ID: 24POST1189614). Dipender Gill is supported by the British Heart Foundation Center of Research Excellence (RE/18/4/34215) at Imperial College. Michael G Levin is supported by the Institute for Translational Medicine and Therapeutics of the Perelman School of Medicine at the University of Pennsylvania, the NIH/NHLBI National Research Service Award postdoctoral fellowship (T32HL007843), and the Measey Foundation. Stephen Burgess is supported by the Wellcome Trust (225790/Z/22/Z) and the United Kingdom Research and Innovation Medical Research Council (MC_UU_00002/7, MC_UU_00040/01).

## Author contributions

S.Y., S.M.D., and S.C.L. conceived and designed the study. S.Y., J.C., Y.L., S.A., and X.R. undertook the statistical analyses. S.Y. wrote the first draft of the manuscript. S.Y., J.C., X.R., Y.L., S.A., L.W., F.J., Y.X., M.G.L., B.F.V., D.G., S.B., A.Å., K.M., X.L., S.M.D., and S.C.L. contributed to data interpretation, offered significant intellectual insights to the manuscript, and approved its final version.

## Funding

## Competing interests

S.M.D. receives research support from RenalytixAI and in-kind research support from Novo Nordisk, both outside the scope of the current project. D.G. is the Chief Executive Officer of Sequoia Genetics, a private limited company that works with investors, pharma, biotech, and academia by performing research that leverages genetic data to help inform drug discovery and development. D.G. has financial interests in several biotechnology companies. Other authors declare no competing interests.

## Additional information

[1]Department of Surgery, University of Pennsylvania Perelman School of Medicine, Philadelphia, PA, USA. [2]Unit of Cardiovascular and Nutritional Epidemiology, Institute of Environmental Medicine, Karolinska Institutet Stockholm, Sweden. [3]Department of Medical Epidemiology and Biostatistics, Karolinska Institutet, Stockholm, Sweden. [4]Corporal Michael J. Crescenz VA Medical Center, Philadelphia, PA, USA. [5]Department of Gastroenterology, Central South University Third Xiangya Hospital, Changsha, Hunan, China. [6]Xiangya School of Public Health, Central South University, Changsha, Hunan, China. [7]School of Public Health, Zhejiang University School of Medicine, Hangzhou, Zhejiang, China. [8]Division of Cardiovascular Medicine, Perelman School of Medicine, University of Pennsylvania, Philadelphia, PA, USA. [9]Department of Genetics, University of Pennsylvania, Philadelphia, PA, USA. [10]Department of Systems Pharmacology and Translational Therapeutics, University of Pennsylvania, Philadelphia, PA, USA. [11]Institute for Translational Medicine and Therapeutics, University of Pennsylvania, Philadelphia, PA, USA. [12]Department of Epidemiology and Biostatistics, School of Public Health, Imperial College London, London, United Kingdom. [13]MRC Biostatistics Unit, University of Cambridge, Cambridge, UK. [14]Department of Public Health and Primary Care, University of Cambridge, Cambridge, UK. [15]Medical Epidemiology, Department of Surgical Sciences, Uppsala University, Uppsala, Sweden. [16]These authors contributed equally: Shuai Yuan, Jie Chen. [17]These authors jointly supervised this work: Scott M. Damrauer, Susanna C. Larsson. ✉e-mail: Shuai.Yuan@Pennmedicine.upenn.edu

