## [Transparent Peer Review file · Nature Communications]

Cross-population GWAS and proteomics improve risk prediction and reveal mechanisms in atrial fibrillation

Corresponding Author: Dr Shuai Yuan

Version 0:

Reviewer comments:

Reviewer #1

(Remarks to the Author)

Overall:

Thank you for the opportunity to review this interesting manuscript by Yuan S and colleagues on a genome-wide association analysis of atrial fibrillation with several downstream in silico analyses to better elucidate mechanisms and advance risk prediction. While this is of potential interest, there are important concerns regarding experimental design leading to challenges in novelty interpretability, particularly in comparison to Miyazawa et al Nat Genet 2023. The TWAS and MR analyses with cardiometabolic traits appear to recapitulate the same findings. While a direct comparison was not performed, AF PRS performance in Miyazawa et al appears greater than the AF PRS presented here.

Major:

1. Miyazawa et al Nat Genet 2023 meta-analyzed prior European AF GWAS, FinnGen v2, and Biobank Japan. However, the present paper meta-analyzed part of the same prior European AF GWAS (without UK Biobank) and FinnGen v10 as a part of discovery then used a meta-analysis of UK Biobank and SIMPLER (from Sweden) for replication. The use of a discovery dataset that is largely a subset of prior AF GWAS does not seem logical.
2. The criteria for ascribing novelty is not described (e.g., genomic window, LD r^2). Additionally, it is not clear whether the authors considered Miyazawa et al Nat Genet 2023 in their classification for novelty.
3. TWAS analyses of cardiac tissue using GTEx was also used in Miyazawa et al Nat Genet 2023 but it is unclear if there is a novel insight in the present TWAS analysis using the same tissue from the same dataset.
4. The PRS analyses would be most informative if benchmarked to other PRS. For example, in Miyazawa et al Nat Genet 2023, the AF PRS led to an AUC of 0.738. However, the present PRS AUC was 0.635. Therefore, it is unclear if the present PRS is an advance.
5. The PRS and MR analyses of AF with other CV traits and risk factors presented also appear to recapitulate observations presented in Miyazawa et al Nat Genet 2023.
6. More details about the proteomics is needed. The main methods indicate that 1817 proteins were analyzed but the supplementary method indicates 4907 from deCODE. Both SomaScan and Olink proteomics have been performed in deCODE (Eldjarn et al Nature 2023). Unfortunately, for a very large fraction of proteins, there is poor agreement across these platforms. This leads to concerns regarding the fidelity of the associations proposed in the proteomics analysis.
7. Approximately 5% of the proteins analyzed showed significant genetic association for AF. Given the high fraction, there is concern about inflation leading to false positives.

Minor:

1. Prior investigations have shown that coding variants may alter epitope binding of aptamer-based proteomic analyses. Sensitivity analyses excluding coding variants would help.
2. The proteomic analyses would be strengthened if the protein concentrations were associated with incident AF.

Reviewer #2

(Remarks to the Author)

Reviewer's comments on "Genome-wide association and Mendelian randomization analyses uncovered 28 novel genetic loci, major modifiable risk factors, and protein therapeutic targets for atrial fibrillation" by Yuan et al.

The authors describe a GWAS meta-analysis of atrial fibrillation (AF), identifying 28 novel loci, followed by downstream analyses including gene prioritization, PRS-analyses, genetic correlation with other phenotypes and two separate Mendelian randomization analyses, one using potential AF risk factors as exposures and one using plasma protein levels. The study adds to loci associated with AF, although the discovery GWAS is largely based on data that was published in 2018 and does not include several new publicly available datasets. My main concerns regard incomplete descriptions of the methodology and presentation of the results, which impede interpretation of the results and conclusions drawn from them. In particular this applies to conclusions about causality drawn from the proteome-wide mendelian randomization analysis. My concerns and questions are outlined in detail in the following comments.

1. If correctly understood, the authors use only publicly available summary statistics for the discovery GWAS, unless the authors have access to individual level data for the SIMPLER cohort? This is not clear from the methods section but should be stated clearly. The sample sizes given for HUNT, deCODE, MGI, discoverEHR, AFgen and UK Biobank in the discovery GWAS match the sample sizes from the 2018 AF GWAS by Nielsen et al. (PMID: 30061737). Thus, the authors seem to have used the summary statistics of the 2018 Nielsen study and added publicly available data from FinnGen and SIMPLER. If correct, this should be stated more clearly. As the authors state, over 100 loci have been identified in previous AF GWASs. It can be argued that at this point, a more comprehensive approach would be appropriate to explore the genetics of AF, e.g. why not include updated UK Biobank data of over 33,000 AF cases instead of summary statistics from 2018 for discovery (N cases = 14820)? The authors have access to this data and use it for testing the AF PRS. The publicly available Million veteran program dataset could also have been included. In this regard, some of the loci reported here as novel, e.g. at LPA, reach GW-significance in the MVP data alone (https://phenomics.va.ornl.gov/pheweb/gia/meta/pheno/Phe_427_2)

2. The authors mention lack of ancestry diversity as a limitation of the study. This is a very important point. AF has been extensively explored in Europeans hitherto. Lack of diversity in genetic studies diminishes power to discover new loci and contributes to health disparities as it limits transferability of the results for risk prediction in other ancestries (PMID: 30901543). In this regard, why did the authors not include publicly available datasets with AF subjects of ancestries other than European in their GWAS (e.g. Million veteran program, All of us, Japan biobank etc.)?

3. The description of the methodology of gene prioritization is insufficient and it is not clear from the text or tables which genes are prioritized. Examples:

a. The overall approach to gene prioritization is not explained – what was the criteria for a gene to be prioritized? What amount of evidence was required?

b. What does “...analysis of coding variants using gene biotypes from the Ensembl database” mean? It is not clear from this description how coding variants were used to prioritize genes. Did the lead AF variants have to be coding or in LD with coding variants? Was it sufficient for the prioritization that a coding variant was in an “independent significant genomic risk loci”, defined earlier in the methods section?

c. The same applies to “eQTL mapping” – how exactly were genes prioritized based on eQTLs? The authors mention “eQTL mapping” but give no further detail on what that entails (e.g. R2 between top AF variant and top-eQTL? Colocalization of AF and eQTL signal?).

d. Was the TWAS a part of the gene prioritization approach?

e. It is not clear what Tables S4 and S5 depict. E.g. Table S4, is only named “Gene prioritization” and no further explanation is given. Abbreviations and column names are not explained, e.g. “num_sig_eqtl”, “num_sig_pqtl”? What does “coding_variant” mean? That the lead variant is a coding variant or that the locus contains a coding variant in the gene? Why are gene names not given, only Ensembl identifications? Why are some loci not included in the table? Likewise, the name of Table S5 is not informing (“Loci gene expression in target tissues”), what is the difference between rsID and LeadSNPs? Are LeadSNPs top eQTLs? This needs explaining and it would be relevant to give R2 between AF-lead SNPs and top-eQTLs. For many of the loci in Table S5, all P-values are >0.05, what was the threshold used to prioritize genes based on eQTLs?

4. Genetic correlation and PRS Phe-WAS: The purpose of these analyses is unclear and the interpretation lacks insight and references to previous studies. Shared genetics between AF and other heart diseases, such as CAD, heart failure (HF), systolic blood pressure and cardiomyopathy have been reported by previous studies (e.g. PMID: 29892015, PMID: 30061737). In fact, Nielsen et al. (on which large part of the GWAS discovery here is based) observed similar associations of AF PRS with cardiovascular phenotypes and concluded that the associations were potentially mediated by assessment bias or the fact that AF may be an intermediate step towards the disease, after removing participants diagnosed with arrhythmia from the analysis (PMID: 30061737). The bidirectional causal relationship between AF and HF has been described in epidemiological (PMID: 26746177) and genetic (PMID: 31919418) studies and AF is a known consequence of cardiomyopathies (PMID: 35165560). It is therefore not appropriate to imply that the results of the current study per se inform clinical decision making as suggested by the following statement in Discussion: “Both genetic correlation analysis and PRS-PheWAS identified genetic liability to AF associated with other heart diseases, like heart failure and cardiomyopathy. These data revealed potential shared genetic mechanisms and, more importantly, suggested that enhanced medical interventions should be made to screen and treat AF among the patients with these heart issues.”

5. Mendelian randomization of 26 modifiable risk factors:

a. A rationalization for the choice of the 26 phenotypes as exposures for the Mendelian randomization analysis is lacking – is there epidemiological evidence or biological plausibility for a role of all phenotypes in the pathophysiology of AF?

b. How do the authors justify using an FDR instead of Bonferroni correction for multiple testing when claiming causality in MR for the 26 modifiable risk factors?

c. Presentation of results in figure 6 is insufficient. In Fig 6a only FDR P-values are given and the ORs and CIs are only shown in figure and not spelled out. Please give P-values, ORs and CIs, as well as number of variants used as instruments and sample sizes for exposure and outcome variables for the primary analysis in a supplementary table.

d. For the risk factors that are claimed to be causal in AF, plots of the instruments' effects on the exposure against their effects on AF should be shown.

6. The proteome-wide MR-analysis needs to be explained better, the methodology and presentation of results is unclear.

a. It is unclear which analysis is used to identify instruments for the MR. The authors state "For circulating proteins, we used lead cis-SNPs associated with the levels of plasma proteins at $P < 5 \times 10^{-8}$ as instrumental variables." – Lead SNPs from which source? No reference is given for this statement. In the Colocalization chapter in the supplement the authors cite Ferkingstad et al. (PMID: 34857953), is this the source for the primary analysis, or the Fenland study (mentioned in relation to sensitivity analysis)? In the main manuscript, the authors only refer to "deCODE protein data" without giving a reference and report a URL directing to a cite with all deCODE summary statistics (<https://www.decode.com/summarydata/>). The figure (Fig7) and table (ST11) presenting the results do not clearly refer to data sources or sample sizes. E.g. ST11 is named "Results of proteome-wide MR analysis of AF" and no further explanation is given (i.e. data source for protein measurement and AF, sample sizes).

b. It is also unclear on which data the sensitivity analyses are based and in which table/figure the results are presented, e.g. in Fig. 7b, what does deCODE_Meta, deCODE_Dis and deCODE_Rep stand for?

c. Why do the authors not use the largest available GAWs of the proteome, based on over 50,000 UKB participants (PMID: 37794188)?

d. In the supplementary methods the authors state that the threshold for colocalization was $PH4 \geq 0.8$, what does the dotted line at approximately 0.7 on Fig 7c represent?

e. "Genetically predicted levels of 86 circulating proteins were associated with AF risk after multiple testing corrections in the discovery analysis (Figure 6a and Table S11)." – This paragraph seems to wrongly refer to Figure 6 instead of Figure 7.

7. Several aspects of the proteome-wide MR can be criticized (the following points are based on my understanding of the incompletely explained approach, see previous comment).

a. Important information about the protein MR-analysis is missing. It is vital to give the number of variants underlying the protein MR-analysis for each protein, information on pleiotropy (do the variants associate with many other proteins or phenotypes?) and show effect-effect plots, at least for protein-AF associations that are being claimed as potentially causal and supported by colocalization. As an example, it would be highly questionable to claim causality for a protein-AF association supported by one or few pleiotropic variants.

b. The authors state that "Independent SNPs were used as an instrumental variable after pruning these SNPs at $r^2 < 0.01$ to minimize the effect of collinearity of selected SNPs in linkage disequilibrium." But it is unclear whether this only applies to the modifiable risk factor analysis or also the proteome-wide MR, as this is stated before introducing the proteome-wide analysis. Since the authors only use cis-instruments for the MR-analysis, I assume that many of them are based on only one or a few instruments, particularly if they apply the $r^2 < 0.01$ threshold, is that correct? In any case, information on the instruments used is not given but is crucial. Using one or few instruments is problematic, e.g. because of the potential of drawing conclusions on causality from one pleiotropic variant and using correlated SNPs as instruments can result in inflated association results.

c. The potential for mediation through AF risk factors is not sufficiently addressed, e.g. in the case of PCSK9, levels of which are strongly associated with CAD (which its inhibition is intended to prevent) and other cardiovascular diseases, the same applies to AGT.

d. The way in which the authors represent the results is not consistent with threshold for colocalization being $PH4 \geq 0.8$. E.g. according to Fig. 7c, only around 10 proteins have evidence above this threshold (which is not the same as the dotted line on the figure), still the authors state in the abstract "Proteome-wide Mendelian randomization and colocalization analyses identified 13 blood proteins (including PCSK9 and AGT as drug targets) with potentially causal roles in AF". Furthermore, in the results, the authors state: "Per SD increase in genetically predicted protein levels, the OR of AF ranged from 0.57 (95% CI 0.49-0.6) for PMVK (Phosphomevalonate Kinase) to 2.57 (95% CI 2.13-3.10) for RAB1A (Ras-related protein Rab-1A)" – Neither of these protein-AF associations are supported by colocalization and therefore this presentation is not appropriate.

e. How is colocalization performed and interpreted when more than one variant is used as an instrument?

f. "For traits with SNPs ≤ 3 , we used the Wald ratio method or the inverse variance weighted method under the fixed effect model to estimate the MR association with AF. Otherwise, the inverse variance weighted method under the multiplicative random effects model was used" – it is not clear for which traits this applies

g. Can the authors justify use of FDR instead of Bonferroni correction for multiple testing? It can be argued that a more conservative method (i.e. Bonferroni) is more appropriate when claiming causality.

h. Figure 7 is confusing and getting information on one protein is difficult. Suggest giving complete information for each protein in table, including evidence from colocalization, number of variants used as instruments in MR and which variants are used as instruments.

8. In general, tables and figures are incompletely described. A few examples have been given already, e.g. see Comment #3e regarding ST4 and ST5. Additional examples:

a. ST2, describing the discovery GWAS: Data sources and sample size are not stated. The columns are incompletely explained, e.g. for genomic locus 170, three different genes are given in the columns "ensemble" (ENSG00000289950), "hgnc" (EMC10) and "symbol" (MYH14) and the meaning of each column is not explained.

b. ST9 is referred to in the text in relation to sensitivity analysis ("The associations remained consistent in sensitivity analyses (Table S9).") but the table title does not mention sensitivity analysis, or in fact which data the analysis is based on ("MR analysis of the associations between modifiable factors and AF").

9. Discussion of the results is limited and lacking appropriate context with previous literature. Examples:

a. "Among novel loci, LPA, CDKN2B-AS1, and ZFPM2 have potential mechanistic support" – are these genes prioritized candidate causal genes at these loci or is this based on proximity only?

- b. "The association between lipoprotein(a) levels and AF may be mediated by valve stenosis, coronary artery disease and heart failure."- this can be further examined, e.g. by comparing the variants effect on these phenotypes with their effects on AF.
- c. Paragraph starting with "Besides individual loci, our pathway analysis identified muscle development and heart contraction as significant contributors to AF. development. ..." – no statements in this paragraph are supported by references. Furthermore, the role of structural remodeling in AF pathogenesis has been highlighted in previous GWASs of AF and discussed with insight, e.g. in relation to the importance of atrial remodeling in AF pathogenesis, including in the Nielsen study, on which a large part of this GWAS discovery is based (PMID: 30061737).
- d. "Heart contraction pathways play a pivotal role in ensuring the synchronized contraction of cardiac muscle fibers, which is essential for effective blood pumping" – It is difficult to understand the meaning of this sentence and how this knowledge relates to the results of the study. Furthermore, no references are cited.
- e. "However, given that AF is a multifactorial disease, our PRS with R^2 of 5.3% is still respectable for disease risk prediction." - I cannot find this statistic anywhere in the results section (text or tables).

Reviewer #3

(Remarks to the Author)

Yuan et al. performed a large-scale meta-analysis of AF-GWAS including approximately 120,000 cases, identifying 178 loci associated with AF, 28 of which were previously unreported. Following this, genetic correlations between AF and relevant traits were assessed, polygenic risk scores were constructed, and potential risk factors and therapeutic targets were examined through proteome-wide Mendelian randomization analysis. Particularly in MR analysis, colocalization and sensitivity analyses were also performed to support the evidence of the MR results. The authors elaborated on the details in the analytical methods and the manuscript is well written, but it seems to be unclear along several issues and may need some additions.

Major comments;

1. When calculating PRS using the P+T method, multiple parameter values were tested, but for the PRS-CS method, no parameter settings were specified. Although this is not explicitly mentioned in the Methods section, was PRS calculated using the default value for the parameter such as phi in the PRS-CS? Since the P+T calculated PRS under several conditions, wouldn't it be worth trying to set the phi parameter under various conditions as well? Additionally, when performing such parameter tuning, I think it is essential to adjust the parameters using an independent validation cohort and then conduct performance evaluations with a separate test cohort.
2. I think that deCODE proteome data is primarily used in the MR analysis. However, this is not mentioned in the Methods, even not in MR analysis in Supplementary Methods. This is only first mentioned in the colocalization analysis section in the Supplementary Methods. While it might be clear after reviewing all the materials, it would be better to introduce the data used in a more logical order, so that readers can understand it more easily, and provide the details in the supplementary materials.
3. As the authors mentioned, sample overlap can introduce bias into MR results. While the sensitivity analyses were performed to ensure no sample overlap in the MR analysis, I think it would be better not to conduct MR analysis with sample overlap from the beginning and to only present the MR without sample overlap. Additionally, if the pQTL and AF-GWAS signals are strongest in the absence of sample overlap, it might be better to focus on the MR analysis using the Fenland study and the meta-analysis of AF-GWAS as the main analysis and use the MR with deCODE and other meta-GWAS excluding deCODE as a replicate.
4. Depending on the protein, multiple pQTLs may serve as instrument variables in MR analysis. Please provide details on the number of pQTLs for each protein used in the MR analysis. Additionally, when using multiple pQTLs of a given protein as instrumental variables, did the authors perform colocalization analysis between individual pQTLs and AF-GWAS and calculate the posterior probability? If so, did the authors consider the average or maximum posterior probability with more than 0.8 to have evidence of colocalization?
5. In the discussion section, the authors stated that muscle development and heart contraction were significant pathways in the gene-enrichment analysis and played critical roles. However, these two pathways have already been reported in the study by Patrick T. Ellinor et al. published in Nature Genetics in 2018. Therefore, it is necessary to clearly distinguish and explicitly highlight the mechanisms revealed by the identification of new loci in this study from what has already been described in previous GWAS.
6. In the discussion section lines 324-326, the authors mentioned that lipids and diabetes may not be important risk whereas these factors are established ones in terms of clinical practice. If so, could you show more evidence to convince readers?
7. Figure 3. Enrichment analyses were performed using just the top 20 signals. Why 20? If you utilized all positive signals/ all negative signals/ all positive and negative signals, what result will we see?
8. Figure 3. The author utilized GTEx data from the left ventricle and atrial appendage. While I agree that these two organs are associated with AF, how about other organs/ tissues? Stratified LDSC or DEPICT algorithm could point out additional AF-related ones.
9. Figure 4. Now that the GWAS summary statistics of many traits are available, could you comprehensively investigate

genetic correlations with other traits than traditional AF risk factors?

10. Figure 5a. While the performance of PRSs was shown, is there statistical difference between those PRSs? You can test it using bootstrap method or just compare their AUROC using Delong test.

11. In the supplementary methods, it is stated the strong evidence of colocalization if the posterior probability for shared causal variants (PH4) was ≥ 0.8 , but in the main manuscript, it is described as having high support of colocalization with $PP.H4 > 0.7$. Which one is correct? Also, what is the rationale for choosing this threshold?

Minor comments;

1. The figures cited in the Blood proteins and AF section are all labeled as Figure 6, but I think it should be Figure 7.

2. Regarding Figure 3c, the description of the figure legend is insufficient. There is no explanation for the abbreviations BB, CC, and MF, nor for the terms 'count' and 'gene ratio'.

3. Figure 5 is generally of low quality. For example, in Figure 5a, there are multiple bars of similar pseudo R^2 values, making it difficult to see under which condition the pseudo- R^2 is the highest. It would be helpful to reorder them in descending order. Additionally, the X-axis of Figure 5a represents PRS parameters and the number of SNPs, but the font size is small, making it difficult to understand. For the sake of clarity, it might be helpful to use color coding for methods like P+T and PRS-CS, or to display the parameters separately instead of including them as axis text. The label of the Y-axis in Figure 5c is missing. In Figure 5e, the labels for the significant phenotypes overlap and are hard to read. The phenotype categories are color-coded with two colors, but using separate colors for each category would make it easier for readers to understand.

Version 1:

Reviewer comments:

Reviewer #2

(Remarks to the Author)

I thank the authors for their responses to my comments, and those of other reviewers. I have several remaining concerns, outlined below. In particular, there is a large overlap of cases between the current study and a recently published AF GWAS and the approach to claiming novel loci ignores published research (in addition to the newly published one). Furthermore, the authors claim differences in genetic correlation between ancestry groups without performing any statistical comparison. Finally, I stand by my previous criticism of claiming that proteins are causal in AF based on MR with single variant as instrument and not giving that variant and its AF association.

1. Overlap of cases with other AF GWASs

In my previous comments I pointed out that a majority of the discovery dataset was the exact same as the data of a 2018 GWAS meta-analysis of atrial fibrillation by Nielsen et al., i.e. the HUNT, deCODE, MGI, DiscoverEHR, AFGen consortium and UK Biobank datasets. Now the authors have updated UK Biobank (and now use it as a replication study) and added the publicly available MVP dataset to their study, which is by far the greatest addition to the European discovery dataset, as well as adding participants of other ancestries. Combined number of cases in the current study is now almost 170,000. However, since the first revision of the current paper, Roselli et al. have published the largest GWAS of AF hitherto, with over 180,000 cases (<https://pubmed.ncbi.nlm.nih.gov/40050429/>). The Roselli study includes updates of many of the cohorts for which the current study uses summary statistics from Nielsen et al., 2018, i.e. HUNT, deCODE, MGI, DiscoverEHR. In fact, it seems that the Simpler cohort of approximately 8500 individuals is the only cohort in the current study that is not included by Roselli et al. in the same or larger version. Otherwise, if I am not mistaken, the current study is largely a subset of the Roselli-2025 GWAS.

2. Novelty of identified loci

In light of the Roselli-2025 publication (<https://pubmed.ncbi.nlm.nih.gov/40050429/>), claims of novelty need to be revisited. Furthermore, the authors approach to defining novelty is incomplete: „Loci without overlapping coordinates for previously reported genome-wide significant variants ($P < 5 \times 10^{-8}$) from prior studies (Nielsen JB et al., 2018;7 Roselli C et al., 2018;8 Miyazawa K et al., 20239) were annotated as novel.“ – it is unclear why the authors restrict to three studies when many more exist and a simple look-up in the GWAS catalog would improve definition of novel loci. Many loci are claimed as novel but have been reported in GWASs of AF (e.g. at BRWD1: <https://pubmed.ncbi.nlm.nih.gov/39024449/> and PLEC: <https://pmc.ncbi.nlm.nih.gov/articles/PMC5704994/>). Another recently published study by Choi et al (<https://www.nature.com/articles/s41588-025-02074-9>), reporting structural and coding variants in AF, would also need to be taken into account when claiming novelty.

3. Publicly available data

The authors have not answered whether they had access to any individual level data (specifically I asked about Simpler). It would be appropriate to state somewhere in the manuscript if the whole study is based on publicly available summary statistics or not.

4. Gene prioritization

The authors have improved description of methodology for gene prioritization. However, it is questionable to put proximity above f.ex. eQTL and not to focus on eQTLs in appropriate tissues (i.e. the heart). Furthermore, many of the prioritized genes

are only supported by proximity, which is very weak evidence.

5. Correcting for multiple testing in enrichment analysis

Regarding “Enrichment analysis in the Reactome database identified 36 out of 1,131 pathways significantly associated with AF after Bonferroni correction.” A bonferroni correction with 1,131 tests would give a p-value threshold of 4.4×10^{-5} . Please explain why 36 pathways are considered significant when only 5 are below this threshold according to Table S5

6. Comparison of genetic correlation with CVDs between ancestry groups

Regarding “However, Europeans had additional associations with valve disease and cardiomegaly, while African Americans showed stronger links to rheumatic heart disease and cerebral ischemia.” – please provide an objective comparison of genetic correlation of AF and each of these phenotypes among Europeans and African Americans, e.g. is there significant heterogeneity?

Furthermore, the authors cite two studies on the matter: “We observed ancestry-specific differences in genetic correlations between AF and circulatory comorbidities, with Europeans showing stronger associations with valve disease and cardiomegaly, while African Americans had stronger links to rheumatic heart disease and cerebral ischemia. These disparities, also reported in other studies,19,20 may arise from ancestry specific genetic architecture, environmental factors, and variations in AF subtypes” – the studies that are cited are observational, not genetic studies, which would be relevant to mention. Furthermore, these studies do not mention differences in valve disease, cardiomegaly and rheumatic heart disease, as the text indicates. One of them (Magnani et al.) examines racial differences in the outcome of AF on the rates of stroke, heart failure, CHD, and mortality and the other (Essien et al.) examines racial differences in anticoagulation prescription and stroke.

7. Mendelian randomization – correcting for multiple testing

Regarding “Additionally, genetically proxied high-density lipoprotein (HDL) cholesterol levels, cigarettes smoked per day, moderate-to-vigorous physical activity, sleep duration, short sleep time, and blood calcium levels were associated with AF at the nominal significance level.” - would recommend adhering to claiming only associations that are significant after correcting for multiple testing.

8. MR-analysis using cis-pQTLs as instruments to identify causal proteins

In my first review I asked if only one instrument was used for each protein-AF MR analysis. The authors have responded that only one cis-pQTL is used in each case. However, they have not responded to my request to give information on which snp is being used as instrument. Knowing the SNP and its association (OR and P-value) with AF is crucial – are the AF associations at noise level considering the large sample size for AF? And if these proteins are causal in AF, would the cis-pQTLs not come up in the AF GWAS? Furthermore, for three of the identified “causal” proteins, effects on AF in the replication dataset (Fenland) were opposite to that of the discovery – could this indicate that the instruments used are not true AF associations?

9. Discussion:

a. “The PITX2 and ZFHX3 have been recognized as well-known genes associated with AF, our study emphasized these two as shared loci across four ancestry groups and thus further confirms their pivotal role in AF susceptibility.” - this is not novel, these loci have been well established in different ancestries through other studies (e.g. <https://pmc.ncbi.nlm.nih.gov/articles/PMC6136836/> and <https://pmc.ncbi.nlm.nih.gov/articles/PMC9925380/>)

b. Two paragraphs are almost exact replicas of each other, starting with:

“We employed a comprehensive gene prioritization strategy, identifying putative causal genes for 504 loci, providing functional insights into AF pathogenesis....”

“We used a series of methods for gene prioritization and identified putative causal genes for 504 loci, providing functional insights into AF pathogenesis....”

Reviewer #3

(Remarks to the Author)

The authors have responded to my comments appropriately.

Just one thing, 95% CI has been added regarding the performance of PRS, making the numerical difference easier to understand. However, it is still unclear whether the performance has improved statistically significantly.

Reviewer #4

(Remarks to the Author)

This is my initial exposure to the current manuscript as I was asked to review the revision on behalf of a prior reviewer. Therefore, my comments are focused on whether the prior reviewer’s concerns were adequately addressed. In general, the revisions have been comprehensive and additive, and the work is a worthy addition to the literature. I do note a few remaining issues:

Major

1. I note that a large cross-ancestry AF GWAS meta-analysis was recently published in Nature Genetics (PMID 40050429).

The current study has actually somewhat fewer cases (168k vs 180k), although more loci are identified. Referencing this recent paper would require a major overhaul to the paper and text, so I will defer to the Editors and journal policy as to whether any action needs to be taken to this effect.

2. The 'replication' design here seems somewhat of an afterthought, as the replication dataset is then simply incorporated into the European and cross-ancestry meta-analyses which are (I think reasonably) presented as the primary analysis. The authors should consider whether the replication design really adds anything here, or whether streamlining the analysis would increase clarity.

3. The authors do not appear to apply a MAF filter (I just see MAC < 50). Note this is in contrast to the other large AF meta-analyses this study is compared to. This issue should be clarified, and the number/% of significant variants with MAF < 1% should be reported. Were more stringent QC parameters applied to rarer variants? Did rarer variants replicate (if the replication approach is retained)?

4. The gene prioritization approach appears substantially more robust and systematic in this version compared to the original. However, I feel the relevant language (Results: "we identified a single putative causal gene at each of the 504 genome-wide significant loci", Limitations: "we cannot entirely rule out the possibility of misassigning genes to loci") is too strong. The authors employ a robust approach but functional annotation remains imperfect and I would argue that misassignment of at least some genes is almost certainly present and inevitable. I recommend the authors nuance their reporting accordingly. To this end, it would be useful if the authors could report the number (%) of loci at which all available prioritization schemes prioritized the same gene.

5. It would be helpful to state in the Results that the PGS analysis in PMBB represents an evaluation in an independent dataset not used for PGS derivation. The authors should also report the ancestral breakdown of the PMBB dataset used for evaluation.

6. Regarding the protein scores, I do not fully understand why 2 approaches are used (LGBM + LASSO). It's specifically not clear to me the need for the LGBM, which is subjected to several steps prone to overfitting (e.g., ranking based on individual information gain, clustering, sequential forward selection), and unsurprisingly is outperformed by the LASSO. This is another potential area where there is unnecessary complexity. I also feel it is not helpful (for either model) to present results from the training set.

7. The association between genetic liability to AF and lower levels of NTproBNP is potentially surprising and merits further discussion. I suppose the current statement that 'NTproBNP elevation in AF is likely a secondary effect of atrial stretch rather than a primary causal factor' could follow from this observation, it seems like an overly strong and somewhat simplistic interpretation, particularly as other factors (e.g., body mass index) may be at play.

8. Related to point #6 above it would be of interest to perform reverse MR for AF to "risk factors". I cannot tell whether this is an analysis that was in the prior version of this manuscript but removed due to prior author comments. If the latter is the case, then this comment may be ignored at the authors' and editors' discretion.

Version 2:

Reviewer comments:

Reviewer #2

(Remarks to the Author)

I thank the authors for their responses to my comments. I want to respond further to their answer to comment #8 (protein-AF MR analysis):

I thank the authors for providing information on the instruments used in pQTL-MR analysis and their association with AF (although giving the protein name in ST20 would help, instead of just the protein ID). I argue that the association with the outcome is relevant. The AF meta-analysis is highly powered so associations close to nominal significance are scattered throughout the genome. I also want to point out that although the instrument's association with the exposure is well established, the exposure/outcome relationship tested is not based on biological plausibility. Thus, the authors are not testing a restricted set of biologically plausible causal factors, they are in fact performing a hypothesis free search for significant MR-analyses using 2,847 cis-pQTLs as instruments, and provide no reference to biological plausibility of a causal relationship between the proteins and AF. Therefore, it could be argued that it would be appropriate to apply a P-value threshold corrected for multiple testing to claim significance for the pQTL-MR analyses.

Reviewer #4

(Remarks to the Author)

Most of my comments have been well-addressed. Just one remaining issue on my end.

I still think the NTproBNP MR findings are not sufficiently handled. Looking at ST15 (UKB) there is an association between genetic liability to AF and lower levels of NTproBNP as well as lower levels of NPPB (NTproBNP precursor). In ST14 (deCODE), there is an association between genetic liability to AF and higher levels of NPPB (and interestingly no representation of NTproBNP at all). By the way, in ST14 there are two entries for NPPB (rows 4 and 13), with different effect sizes, both positive, and I'm not sure what to make of that.

The association of genetic liability to AF and lower levels of NTproBNP is correctly cited in the Results, but in the Discussion the authors seem to write the opposite: "genetic liability to AF was associated with increased levels of NTproBNP, consistent with several observational studies." I believe there is some confusion here. The authors should revise the discussion to align with the results, which are not immediately consistent with the observational findings (which is what

prompted my comment in the first place). Further, the conflicting associations for NPPB are also probably worth mentioning. In sum though, I do generally agree with the authors interpretation that the evidence here does not support a causal role for AF on observed NTproBNP elevations in AF, but that relations are likely complex, potentially bidirectional, and require further investigation.

Reviewer #1:

Thank you for the opportunity to review this interesting manuscript by Yuan S and colleagues on a genome-wide association analysis of atrial fibrillation with several downstream in silico analyses to better elucidate mechanisms and advance risk prediction. While this is of potential interest, there are important concerns regarding experimental design leading to challenges in novelty interpretability, particularly in comparison to Miyazawa et al Nat Genet 2023. The TWAS and MR analyses with cardiometabolic traits appear to recapitulate the same findings. While a direct comparison was not performed, AF PRS performance in Miyazawa et al appears greater than the AF PRS presented here.

Response to the comment: We sincerely thank the Reviewer for their thorough assessment of our paper and for providing insightful comments. We have now updated the analysis by including data from extra studies. Compared to the study by Miyazawa et al. (Nat Genet, 2023) with 77,690 cases, our GWAS includes more than twice the number of cases (168,007) from five different ancestries, offering enhanced statistical power for both MR and TWAS analyses. Compared with previous large-scale GWASs, our study identified over 300 unreported loci and causal circulating proteins, shedding light on the genetic etiology of the disease and highlighting potential therapeutic targets. We have also incorporated polygenic and protein scores for disease risk prediction. Overall, the revision has significantly improved the paper. We have addressed the Reviewer's comments in detail below and revised the manuscript accordingly.

Major:

1. Miyazawa et al Nat Genet 2023 meta-analyzed prior European AF GWAS, FinnGen v2, and Biobank Japan. However, the present paper meta-analyzed part of the same prior European AF GWAS (without UK Biobank) and FinnGen v10 as a part of discovery then used a meta-analysis of UK Biobank and SIMPLER (from Sweden) for replication. The use of a discovery dataset that is largely a subset of prior AF GWAS does not seem logical.

Response to comment 1: Thank you for raising this point. In the updated analysis, we included data from eight studies—HUNT, deCODE, MGI, DiscoverEHR, AFGen consortium, FinnGen R12, MVP, and SIMPLER—in the discovery dataset, resulting in a sample size of 117,905 cases and 1,239,541 controls. This dataset was subsequently meta-analyzed with the replication study (UK Biobank), which contributed 36,075 cases and 371,874 controls. The final European GWAS sample size thus comprises 153,980 cases and 1,611,415 controls, providing robust statistical power for our analysis. Additionally, we integrated data from diverse ancestries to enhance ancestry-specific association detection and improve our findings' generalizability.

2. The criteria for ascribing novelty is not described (e.g., genomic window, LD r2). Additionally, it is not clear whether the authors considered Miyazawa et al Nat Genet 2023 in their classification for novelty.

Response to comment 2: Following the locus definition used in Miyazawa et al. (Nat Genet, 2023), we defined a locus as follows: (1) genome-wide significant variants ($P < 5 \times 10^{-8}$) were

extracted from the association results, (2) a 500-kb window was added on both sides of these variants, and (3) overlapping regions were merged. To assess novelty, loci were considered novel if they did not contain SNPs that had previously been reported as genome-wide significant ($P < 5 \times 10^{-8}$) in large-scale AF GWASs, including those by Nielsen JB et al. (2018), Roselli C et al. (2018), and Miyazawa K et al. (2023).

For GWAS including all ancestries, we identified 543 loci, of which 379 were reported novel. For European GWAS, we identified 504 loci, of which 342 were reported novel. We have updated this information in the paper.

3. TWAS analyses of cardiac tissue using GTEx was also used in Miyazawa et al Nat Genet 2023 but it is unclear if there is a novel insight in the present TWAS analysis using the same tissue from the same dataset.

Response to comment 3: Thank you for the insightful comment. Based on this larger GWAS, we first performed MAGMA to assess the enrichment of AF-associated gene expression across 54 tissue. We observed significant enrichment in three heart-related tissues: Heart_Atrial_Appendage, Artery_Tibial, and Heart_Left_Ventricle. We also performed stratified-LDSC using GTEx data as required by other Reviewers and found same signals. Consequently, we conducted TWAS using data from these three tissues.

In comparison, Miyazawa et al. (Nat Genet, 2023) performed TWAS in only two tissues—Heart_Atrial_Appendage and Heart_Left_Ventricle—without including Artery_Tibial. Moreover, our TWAS analysis is statistically more powerful due to the larger GWAS sample size. As a result, we identified a greater number of AF-associated genes after Bonferroni multiple testing correction: 481 in Heart_Atrial_Appendage, 453 in Artery_Tibial, and 468 in Heart_Left_Ventricle. There were 161 genes associated with AF shared by three tissues.

TWAS is now widely regarded as a valuable approach for gene prioritization. We, therefore, have now deemed TWAS as one of the gene prioritization approaches. Among the genes prioritized by TWAS in our study, 170 overlapped with those identified by other gene prioritization methods (e.g., nearest genes, MAGMA, PoPS, eQTL coloc, CARMA coding gene), further supporting their potential role in AF. We have updated the corresponding text and results in the manuscript to reflect these changes.

4. The PRS analyses would be most informative if benchmarked to other PRS. For example, in Miyazawa et al Nat Genet 2023, the AF PRS led to an AUC of 0.738. However, the present PRS AUC was 0.635. Therefore, it is unclear if the present PRS is an advance.

Response to comment 4: Thank you for the comment. We have now derived our polygenic risk score (PGS) based on the cross-ancestry summary data using PRS-CSx method. We then tested the association with prevalent AF and its risk prediction ability in an independent cohort (Penn Medicine Biobank) of our PGS vs. the PGS from the Miyazawa et al study. We found that our PGS showed superior predictive performance compared to PGS002814 from the Miyazawa et al. study, with an AUC of 0.780 (95% CI: 0.778–0.783) and a Brier score of 0.092 (95% CI: 0.091–0.093), outperforming PGS002814 (AUC = 0.767, 95% CI: 0.764–0.769; Brier score = 0.094, 95% CI: 0.093–0.095). We have updated this part of the paper.

5. The PRS and MR analyses of AF with other CV traits and risk factors presented also appear to recapitulate observations presented in Miyazawa et al Nat Genet 2023.

Response to comment 5: Because the analysis in Miyazawa et al., Nat Genet 2023, overlaps with this part of the study, we excluded it to ensure novelty and avoid redundancy.

6. More details about the proteomics is needed. The main methods indicate that 1817 proteins were analyzed but the supplementary method indicates 4907 from deCODE. Both SomaScan and Olink proteomics have been performed in deCODE (Eldjarn et al Nature 2023). Unfortunately, for a very large fraction of proteins, there is poor agreement across these platforms. This leads to concerns regarding the fidelity of the associations proposed in the proteomics analysis.

Response to comment 6: We thank the Reviewer for the insightful comment. We agree that there is moderate consistency between SomaScan and Olink in protein measurement. However, studies by Eldjarn et al. (Nature, 2023) and Pietzner et al. (Nature Communications, 2021) have demonstrated greater concordance between matching assays on these platforms when cis pQTLs are detected on both.

In the revised version, we thus performed Mendelian randomization (MR) analyses to assess the associations between circulating proteins and atrial fibrillation (AF) risk using cis pQTL data from the deCODE and UKB-PPP studies. Given that Olink identified more cis pQTLs (Eldjarn et al. Nature, 2023) and UKB-PPP had a larger sample size, we prioritized UKB-PPP data for proteins present in both studies. We finally included 2,847 proteins with cis pQTL in the analysis.

To ensure homogeneity, we evaluated the consistency of protein-AF MR associations for the 915 proteins with cis pQTLs available in both datasets. After Bonferroni correction, 96% of the associations retained consistent significance levels. Regarding directionality, 114 out of 128 associations with a p -value < 0.05 in both deCODE and UKB-PPP datasets shared the same direction (Figure 1a). Among the 25 proteins significantly associated with AF in both deCODE and UKB-PPP after multiple testing correction, 22 associations were directionally consistent (**Figure 1b**).

These results demonstrate high consistency in MR evidence based on cis pQTLs from the two platforms, supporting the robustness and reliability of our analytic strategy. We have thus updated the corresponding information in the manuscript.

Figure 1. Directional consistency of protein-AF associations using *cis* pQTL data from deCODE and UKB-PPP.

7. Approximately 5% of the proteins analyzed showed significant genetic association for AF. Given the high fraction, there is concern about inflation leading to false positives.

Response to comment 7: Thank you for the valuable comment. We updated our analytical approach for the protein-AF analysis, transitioning from *cis* two-sample MR analysis to SMR analysis, as SMR (Zhu et al., Nature Genetics, 2016) incorporates the HEIDI test to detect pleiotropy that could lead to false positives. We also implemented genetic colocalization analysis to strengthen causal inference further and exclude false positives caused by linkage disequilibrium.

In summary, we investigated the associations between genetically predicted levels of 2,847 proteins and AF risk using data from UKB-PPP and deCODE. After applying rigorous filtering criteria, including Bonferroni-corrected association P -values ($P < 0.05$), HEIDI test results ($P > 0.05$), and strong colocalization evidence ($\text{PPH4} \geq 0.8$), we identified genetically predicted levels of 28 proteins that were significantly associated with AF risk.

Minor:

8. Prior investigations have shown that coding variants may alter epitope binding of aptamer-based proteomic analyses. Sensitivity analyses excluding coding variants would help.

Response to comment 8: Thank you for this insightful comment. We agree that including variants altering epitope binding of aptamer-based proteomic analyses can influence MR results. However, it is not clear what proteins with coding variants may alter the epitope binding of aptamer-based proteomic analyses. We have discussed this as a limitation in the paper.

Page 14: “Fourth, the inclusion of coding variants may alter epitope binding in aptamer-based proteomic analyses for certain proteins, potentially introducing measurement bias that could affect the accuracy of MR results.⁴²”

9. The proteomic analyses would be strengthened if the protein concentrations were associated with incident AF.

Response to comment 9: We appreciate the Reviewer’s suggestion that incorporating evidence from different study designs can enhance the robustness of our findings. However, accurately defining covariates in cohort studies poses methodological challenges, as residual confounding may still persist despite careful adjustments. Therefore, we adhered to the Mendelian randomization framework for causal inference on the association between proteins and AF risk, leveraging different protein data sources for replication. Furthermore, we expanded our analysis to include AF risk prediction, integrating polygenic risk scores and protein scores. To identify proteins predictive of incident AF, we applied the LightGBM method, first selecting proteins associated with incident AF using Cox regression. The full results of the Cox regression analysis are provided in Table S19.

Reviewer #2:

The authors describe a GWAS meta-analysis of atrial fibrillation (AF), identifying 28 novel loci, followed by downstream analyses including gene prioritization, PRS-analyses, genetic correlation with other phenotypes and two separate Mendelian randomization analyses, one using potential AF risk factors as exposures and one using plasma protein levels. The study adds to loci associated with AF, although the discovery GWAS is largely based on data that was published in 2018 and does not include several new publicly available datasets. My main concerns regard incomplete descriptions of the methodology and presentation of the results, which impede interpretation of the results and conclusions drawn from them. In particular this applies to conclusions about causality drawn from the proteome-wide mendelian randomization analysis. My concerns and questions are outlined in detail in the following comments.

Response to the comment: We sincerely thank the Reviewer for the thorough evaluation of our manuscript and for providing insightful comments. In this revised version, we incorporated additional data from MVP, updated GWAS in UK Biobank, updated FinnGen, Biobank Japan, and Genes & Health, resulting in a more than twofold increase in the number of cases compared to Miyazawa et al. study (Nature Genetics, 2023). Building on this comprehensive GWAS meta-analysis, our study identified over 300 unreported loci, offering valuable insights into the genetic etiology of the disease. We have refined our analytical strategy for the protein-AF MR analysis to improve robustness. As recommended, we have included a more detailed description of the methodology.

1. If correctly understood, the authors use only publicly available summary statistics for the discovery GWAS, unless the authors have access to individual level data for the SIMPLER cohort? This is not clear from the methods section but should be stated clearly. The sample sizes given for HUNT, deCODE, MGI, discoverEHR, AFgen and UK Biobank in the discovery GWAS match the sample sizes from the 2018 AF GWAS by Nielsen et al. (PMID: 30061737). Thus, the authors seem to have used the summary statistics of the 2018 Nielsen study and added publicly available data from FinnGen and SIMPLER. If correct, this should be stated more clearly. As the authors state, over 100 loci have been identified in previous AF GWASs. It can be argued that at this point, a more comprehensive approach would be appropriate to explore the genetics of AF, e.g. why not include updated UK Biobank data of over 33,000 AF cases instead of summary statistics from 2018 for discovery (N cases = 14820)? The authors have access to this data and use it for testing the AF PRS. The publicly available Million veteran program dataset could also have been included. In this regard, some of the loci reported here as novel, e.g at LPA, reach GW-significance in the MVP data alone

(https://phenomics.va.ornl.gov/pheweb/gia/meta/pheno/Phe_427_2)

Response to comment 1: Thank you for the valuable comment. As suggested, we incorporated additional data into the GWAS meta-analysis to enhance the study's scope and statistical power.

For the European GWAS meta-analysis, we included data from HUNT, deCODE, MGI, DiscoverEHR, the AFGen consortium, FinnGen R12, MVP, and SIMPLER in the discovery analysis. For replication, we utilized the updated UK Biobank dataset, which now includes 36,075 AF cases. We included data from Biobank Japan, Genes & Health, and MVP for non-European ancestries. In total, our cross-ancestry meta-analysis comprised 168,007 cases and 1,959,739 controls, identifying 525 genomic loci associated with atrial fibrillation. Notably, compared to previously reported loci in large-scale GWAS studies (Nielsen JB et al., 2018; Roselli C et al., 2018; Miyazawa K et al., 2023), our analysis identified 363 unreported loci, providing deeper insights into the genetic architecture of AF. We have updated the manuscript correspondingly.

Data source	Cases	Controls
European		
Discovery analysis		
HUNT	6493	63142
deCODE	13471	358161
MGI	1226	11049
DiscoverEHR	6679	41803
AFGen consortium	17931	115142
The FinnGen R12	63532	252810
MVP_EUR	74946	369730
SIMPLER	8573	27704
Total discovery	117905	1239541
Replication analysis		
UK Biobank	36075	371874
Total European	153980	1611415
Non-European		
Biobank Japan	9826	140446
Genes & Health	754	52054

MVP_AFR	9485	109006
MVP_AMR	3447	46818
Total non-Europeans	14027	348324
Total participants	168007	1959739

2. The authors mention lack of ancestry diversity as a limitation of the study. This is a very important point. AF has been extensively explored in Europeans hitherto. Lack of diversity in genetic studies diminishes power to discover new loci and contributes to health disparities as it limits transferability of the results for risk prediction in other ancestries (PMID: 30901543). In this regard, why did the authors not include publicly available datasets with AF subjects of ancestries other than European in their GWAS (e.g. Million veteran program, All of us, Japan biobank etc.)?

Response to comment 2: We agree that incorporating data from diverse genetic ancestries enhances the robustness of the study. As suggested, we included data for East Asian populations (EAS) from Biobank Japan, American African (AFR) and Admixed American (AMR) populations from MVP, and South Asian (SAS) populations from Genes & Health. We have updated the manuscript correspondingly.

3. The description of the methodology of gene prioritization is insufficient and it is not clear from the text or tables which genes are prioritized. Examples:

- The overall approach to gene prioritization is not explained – what was the criteria for a gene to be prioritized? What amount of evidence was required?
- What does “....analysis of coding variants using gene biotypes from the Ensembl database” mean? It is not clear from this description how coding variants were used to prioritize genes. Did the lead AF variants have to be coding or in LD with coding variants? Was it sufficient for the prioritization that a coding variant was in an “independent significant genomic risk loci”, defined earlier in the methods section?
- The same applies to “eQTL mapping” – how exactly were genes prioritized based on eQTLs? The authors mention “eQTL mapping” but give no further detail on what that entails (e.g. R2 between top AF variant and top-eQTL? Colocalization of AF and eQTL signal?).
- Was the TWAS a part of the gene prioritization approach?
- It is not clear what Tables S4 and S5 depict. E.g. Table S4, is only named “Gene prioritization” and no further explanation is given. Abbreviations and column names are not explained, e.g. “num_sig_eqtl”, “num_sig_pqtl”? What does “coding_variant” mean? That the lead variant is a coding variant or that the locus contains a coding variant in the gene? Why are gene names not given, only Ensembl identifications? Why are some loci not included in the table? Likewise, the name of Table S5 is not informing (“Loci gene expression in target tissues”), what is the difference between rsID and LeadSNPs? Are LeadSNPs top eQTLs? This needs explaining and it would be relevant to give R2 between AF-lead SNPs and top-eQTLs. For many of the loci in Table S5, all P-values are >0.05, what was the threshold used to prioritize genes based on eQTLs?

Response to comment 3: Thank you for your valuable comment. We have re-conducted gene prioritization using updated GWAS data. In this revised analysis, we employed six gene

prioritization approaches: 1) Nearest Gene; 2) MAGMA; 3) PoPS; 4) eQTL colocalization against eQTLGen; 5) encoding variants in credible sets identified by CARMA; and 6) TWAS. For each locus, the prioritized gene was determined by selecting the gene with the highest count of selections across the six prioritization methods. In cases where multiple genes had equal counts, we prioritized genes based on encoding variants in credible sets identified by CARMA, followed by the nearest gene. We have provided detailed descriptions of each gene prioritization method and the corresponding results in the manuscript to ensure clarity and reproducibility.

4. Genetic correlation and PRS Phe-WAS: The purpose of these analyses is unclear and the interpretation lacks insight and references to previous studies. Shared genetics between AF and other heart diseases, such as CAD, heart failure (HF), systolic blood pressure and cardiomyopathy have been reported by previous studies (e.g. PMID: 29892015, PMID: 30061737). In fact, Nielsen et al. (on which large part of the GWAS discovery here is based) observed similar associations of AF PRS with cardiovascular phenotypes and concluded that the associations were potentially mediated by assessment bias or the fact that AF may be an intermediate step towards the disease, after removing participants diagnosed with arrhythmia from the analysis (PMID: 30061737). The bidirectional causal relationship between AF and HF has been described in epidemiological (PMID: 26746177) and genetic (PMID: 31919418) studies and AF is a known consequence of cardiomyopathies (PMID: 35165560). It is therefore not appropriate to imply that the results of the current study per se inform clinical decision making as suggested by the following statement in Discussion: “Both genetic correlation analysis and PRS-PheWAS identified genetic liability to AF associated with other heart diseases, like heart failure and cardiomyopathy. These data revealed potential shared genetic mechanisms and, more importantly, suggested that enhanced medical interventions should be made to screen and treat AF among the patients with these heart issues.”

Response to comment 4: We have removed the hypothesis-free PRS-PheWAS analysis and instead conducted a comprehensive genetic correlation analysis between atrial fibrillation (AF) and near 130 circulatory system outcomes using MVP data, separately in European and African American populations, to identify comorbidities between ancestries. After Bonferroni correction, we identified significant 95 genetic correlations in Europeans, compared to 18 in African Americans. Notably, the top correlations differed substantially between the two populations, suggesting ancestry-specific disease clusters associated with AF.

These findings underscore the importance of considering genetic ancestry in studying disease comorbidities. The differences in correlation patterns may reflect variations in genetic architecture, environmental exposures, or healthcare disparities across populations. From a clinical perspective, these results highlight the need for tailored approaches to disease prevention, diagnosis, and management in diverse populations. For example, understanding ancestry-specific disease clusters could inform targeted screening strategies for comorbid conditions in patients with AF, potentially improving early detection and outcomes. However, we acknowledge that the analysis of African Americans may be underpowered due to this population's smaller number of cases. We have updated the corresponding text in the paper.

5. Mendelian randomization of 26 modifiable risk factors:

- a. A rationalization for the choice of the 26 phenotypes as exposures for the Mendelian randomization analysis is lacking – is there epidemiological evidence or biological plausibility for a role of all phenotypes in the pathophysiology of AF?
- b. How do the authors justify using an FDR instead of Bonferroni correction for multiple testing when claiming causality in MR for the 26 modifiable risk factors?
- c. Presentation of results in figure 6 is insufficient. In Fig 6a only FDR P-values are given and the ORs and CIs are only shown in figure and not spelled out. Please give P-values, ORs and CIs, as well as number of variants used as instruments and sample sizes for exposure and outcome variables for the primary analysis in a supplementary table.
- d. For the risk factors that are claimed to be causal in AF, plots of the instruments' effects on the exposure against their effects on AF should be shown.

Response to comment 5: Thank you for your valuable suggestion. As recommended, we revisited and reselected phenotypes as exposures based on a recent comprehensive review on AF (Elliott et al., Nat Rev Cardiol, 2023). We now focus on 37 modifiable exposures spanning several categories, including adiposity, blood lipids, type 2 diabetes and glycemic traits, other metabolic traits (i.e., blood pressure, thyroid function, and kidney function), lifestyle factors (i.e., smoking, alcohol and coffee consumption, and physical activity), sleep, and dietary factors (i.e., circulating levels of vitamins and minerals).

We used the inverse variance weighted (IVW) method with a fixed-effects model for exposures with fewer than five genetic instruments. For exposures with five or more genetic instruments, we employed MR-PRESSO as the primary analysis method since the method can provide estimates after the removal of pleiotropic SNPs. Notably, MR-PRESSO provides estimates equivalent to the IVW method without outlier SNPs exhibiting pleiotropic effects. Several sensitivity analyses were employed, including the IVW method with random effects, the weighted median, and MR-Egger. We applied Bonferroni correction instead of FDR correction to minimize the false positive rate. We found that genetically predicted BMI, waist-to-hip ratio, visceral adiposity, childhood BMI, apolipoprotein A-I, apolipoprotein B, low-density-lipoprotein cholesterol, type 2 diabetes, systolic and diastolic blood pressure, thyroid-stimulating hormone, smoking initiation, lifetime smoking index, alcohol consumption, leisure screen time, and insomnia were associated with AF risk.

The results, including association estimates, sensitivity analyses, and the number of instruments, are presented in Table S9. For exposures showing significant associations, we created scatter plots to visualize potential pleiotropy and assess the robustness of the findings. Results are displayed in Figure S2-S17.

6. The proteome-wide MR-analysis needs to be explained better, the methodology and presentation of results is unclear.

- a. It is unclear which analysis is used to identify instruments for the MR. The authors state “For circulating proteins, we used lead cis-SNPs associated with the levels of plasma proteins at $P < 5 \times 10^{-8}$ as instrumental variables.” – Lead SNPs from which source? No reference is given for this statement. In the Colocalization chapter in the supplement the authors cite Ferkingstad et

- a. (PMID: 34857953), is this the source for the primary analysis, or the Fenland study (mentioned in relation to sensitivity analysis)? In the main manuscript, the authors only refer to “deCODE protein data” without giving a reference and report a URL directing to a cite with all deCODE summary statistics (<https://www.decode.com/summarydata/>). The figure (Fig7) and table (ST11) presenting the results do not clearly refer to data sources or sample sizes. E.g. ST11 is named “Results of proteome-wide MR analysis of AF” and no further explanation is given (i.e. data source for protein measurement and AF, sample sizes).
- b. It is also unclear on which data the sensitivity analyses are based and in which table/figure the results are presented, e.g. in Fig. 7b, what does deCODE_Meta, deCODE_Dis and deCODE_Rep stand for?
- c. Why do the authors not use the largest available GAWS of the proteome, based on over 50,000 UKB participants (PMID: 37794188)?
- d. In the supplementary methods the authors state that the threshold for colocalization was $PH4 \geq 0.8$, what does the dotted line at approximately 0.7 on Fig 7c represent?
- e. “Genetically predicted levels of 86 circulating proteins were associated with AF risk after multiple testing corrections in the discovery analysis (Figure 6a and Table S11).” – This paragraph seems to wrongly refer to Figure 6 instead of Figure 7.

Response to comment 6: We thank the Reviewer for this insightful point. We have updated our analytic strategy for the pQTL analysis to comprehensively identify causal proteins associated with AF (**Figure 2**). Specifically, we combined two pQTL datasets (deCODE and UKB-PPP) for instrument selection. After removing overlapping proteins, we included 2,847 proteins with cis-SNPs in the analysis. For proteins present in both datasets, we prioritized UKB-PPP data due to its larger sample size and the fact that Olink identified a greater number of cis pQTLs (Eldjarn et al., Nature, 2023). Notably, the associations with AF for these overlapping proteins demonstrated good consistency between the two datasets (See reply to the comment 6 of Reviewer #1). For overlapping protein panels between the deCODE and Fenland studies, we used the Fenland study as the replication dataset due to its smaller sample size.

Regarding the statistical methodology, we transitioned from a cis two-sample MR analysis to SMR analysis (Zhu et al., Nature Genetics, 2016), incorporating the HEIDI test to detect pleiotropy that could otherwise lead to false positives. Comments from other Reviewers also prompted this change. We implemented genetic colocalization analysis to further strengthen causal inference to rule out false positives driven by linkage disequilibrium. A stringent colocalization threshold ($PPH4 \geq 0.8$) was applied to ensure robust findings.

We have updated the corresponding sections on methods and results in the manuscript to reflect these changes and emphasize our approach's methodological rigor.

Figure 2. Study design of Mendelian randomization analysis of association between circulating proteins and AF risk.

7. Several aspects of the proteome-wide MR can be criticized (the following points are based on my understanding of the incompletely explained approach, see previous comment).

Response to comment 7: Thank you for the comment. We have now updated the analytic strategy and completed the methods section of the manuscript, as suggested.

a. Important information about the protein MR-analysis is missing. It is vital to give the number of variants underlying the protein MR-analysis for each protein, information on pleiotropy (do the variants associate with many other proteins or phenotypes?) and show effect-effect plots, at least for protein-AF associations that are being claimed as potentially causal and supported by colocalization. As an example, it would be highly questionable to claim causality for a protein-AF association supported by one or few pleiotropic variants.

Response to comment 7a: Both two-sample MR and SMR, the two most widely used methods, utilize a lead cis-SNP for each protein as the genetic instrument. To minimize the risk of pleiotropy, we transitioned from cis two-sample MR analysis to SMR (Zhu et al., Nature Genetics, 2016), incorporating the HEIDI test to detect potential pleiotropy and reduce false positives. In response to the suggestion, we added an effect-effect plot for proteins that passed filtering based on p-value, HEIDI test, and colocalization test criteria (**Figure 3**). Notably, SNPs with similar effects on AF and pQTL were located on different chromosomes, further supporting the specificity of the associations. Since only cis-SNPs were used as genetic instruments, the likelihood of pleiotropy influencing our findings is minimal.

Figure 3. an effect-effect plot of SNPs against protein levels and AF risk.

b. The authors state that “Independent SNPs were used as an instrumental variable after pruning these SNPs at $r^2 < 0.01$ to minimize the effect of collinearity of selected SNPs in linkage disequilibrium.” But it is unclear whether this only applies to the modifiable risk factor analysis or also the proteome-wide MR, as this is stated before introducing the proteome-wide analysis. Since the authors only use *cis*-instruments for the MR-analysis, I assume that many of them are based on only one or a few instruments, particularly if they apply the $r^2 < 0.01$ threshold, is that correct? In any case, information on the instruments used is not given but is crucial. Using one or few instruments is problematic, e.g. because of the potential of drawing conclusions on causality from one pleiotropic variant and using correlated SNPs as instruments can result in inflated association results.

Response to comment 7b: Yes, the SMR analysis for the protein-AF association is based on a single lead *cis*-SNP. When performing MR analysis using pQTLs, *cis*-pQTLs—those located near the gene encoding the protein of interest—are generally preferred as instrumental variables. This preference is due to their higher specificity and lower risk of pleiotropy compared to *trans*-pQTLs, which are located far from the gene and are more likely to have indirect effects on protein levels. Therefore, selecting *cis*-pQTLs as genetic instruments is an effective strategy to mitigate pleiotropy.

Additionally, SNPs in the *cis* region are often highly correlated due to linkage disequilibrium (LD). To account for this, correlated SNPs should be clumped using a standardized criterion to ensure that only the most representative SNP (the lead *cis*-SNP) is selected. This approach not only simplifies the analysis but also reduces redundancy and the potential for confounding effects. As such, selecting the lead *cis*-SNP for pQTL MR analysis is a

robust and reliable strategy for instrument selection. To minimize pleiotropy and strengthen causal inference, we also used HEIDI and colocalization analysis. We have updated this information in the text.

c. The potential for mediation through AF risk factors is not sufficiently addressed, e.g. in the case of PCSK9, levels of which are strongly associated with CAD (which it is inhibition is intended to prevent) and other cardiovascular diseases, the same applies to AGT.

Response to comment 7c: Based on the current pQTL MR analytic strategy, the associations for PCSK9 and AGT did not persist. While exploring the mediation pathways from proteins to AF is an intriguing area of research, it falls beyond the scope of the present study. Due to too much information derived from our analysis, we decided not to include this in the current study.

d. The way in which the authors represent the results is not consistent with threshold for colocalization being $PH4 \geq 0.8$. E.g. according to Fig. 7c, only around 10 proteins have evidence above this threshold (which is not the same as the dotted line on the figure), still the authors state in the abstract “Proteome-wide Mendelian randomization and colocalization analyses identified 13 blood proteins (including PCSK9 and AGT as drug targets) with potentially causal roles in AF”. Furthermore, in the results, the authors state: “Per SD increase in genetically predicted protein levels, the OR of AF ranged from 0.57 (95% CI 0.49-0.6) for PMVK (Phosphomevalonate Kinase) to 2.57 (95% CI 2.13-3.10) for RAB1A (Ras-related protein Rab-1A)” – Neither of these protein-AF associations are supported by colocalization and therefore this presentation is not appropriate.

Response to comment 7d: We have now applied a stringent threshold for colocalization, setting $PPH4 \geq 0.8$. As suggested, we present the associations that remained significant after filtering based on MR *p*-values, the HEIDI test, and colocalization analysis. Following this approach, we identified 28 associations with causal potential and have updated the corresponding text in the manuscript accordingly.

e. How is colocalization performed and interpreted when more than one variant is used as an instrument?

Response to comment 7e: In the SMR analysis, each protein is represented by a single lead *cis*-SNP as the instrumental variable. For the HEIDI analysis, multiple SNPs (ranging from 3 to 20) were utilized to evaluate potential pleiotropy. In the colocalization analysis, all SNPs within the *cis* region of the encoding gene for each protein were included.

f. “For traits with SNPs ≤ 3 , we used the Wald ratio method or the inverse variance weighted method under the fixed effect model to estimate the MR association with AF. Otherwise, the inverse variance weighted method under the multiplicative random effects model was used” – it is not clear for which traits this applies.

Response to comment 7f: Thank you for the comment. For modifiable risk factors, we have now applied the IVW random-effects method for traits with fewer than 5 SNPs. For traits with 5 or more SNPs, we used MR-PRESSO as the primary analysis method, as it can provide adjusted estimates after removing pleiotropic SNPs. Notably, MR-PRESSO yields estimates equivalent to the IVW method in cases where no outlier SNPs exhibit pleiotropic effects. For the pQTL MR analysis, we utilized the SMR approach to derive association estimates. We have clarified these methodological details in the manuscript as suggested.

g. Can the authors justify use of FDR instead of Bonferroni correction for multiple testing? It can be argued that a more conservative method (i.e. Bonferroni) is more appropriate when claiming causality.

Response to comment 7g: As suggested, we have now used Bonferroni correction for MR analysis of modifiable risk factors and pQTL.

h. Figure 7 is confusing and getting information on one protein is difficult. Suggest giving complete information for each protein in table, including evidence from colocalization, number of variants used as instruments in MR and which variants are used as instruments.

Response to comment 7g: We have revised the figure to enhance clarity and have included the corresponding information in the Table S9 and Table S10 for better reference.

8. In general, tables and figures are incompletely described. A few examples have been given already, e.g. see Comment #3e regarding ST4 and ST5. Additional examples:

- ST2, describing the discovery GWAS: Data sources and sample size are not stated. The columns are incompletely explained, e.g. for genomic locus 170, three different genes are given in the columns “ensemble” (ENSG00000289950), “hgnc” (EMC10) and “symbol” (MYH14) and the meaning of each column is not explained.
- ST9 is referred to in the text in relation to sensitivity analysis (“The associations remained consistent in sensitivity analyses (Table S9).”) but the table title does not mention sensitivity analysis, or in fact which data the analysis is based on (“MR analysis of the associations between modifiable factors and AF”).

Response to comment 8: We apologize for the earlier misunderstanding caused by the information in the tables and figures. We have now thoroughly revised all tables and figures to enhance their clarity and ensure they are more informative.

9. Discussion of the results is limited and lacking appropriate context with previous literature. Examples:

- “Among novel loci, LPA, CDKN2B-AS1, and ZFPM2 have potential mechanistic support” – are these genes prioritized candidate causal genes at these loci or is this based on proximity only?
- “The association between lipoprotein(a) levels and AF may be mediated by valve stenosis, coronary artery disease and heart failure.”- this can be further examined, e.g. by comparing the variants effect on these phenotypes with their effects on AF.

- c. Paragraph starting with “Besides individual loci, our pathway analysis identified muscle development and heart contraction as significant contributors to AF. development. ...” – no statements in this paragraph are supported by references. Furthermore, the role of structural remodeling in AF pathogenesis has been highlighted in previous GWASs of AF and discussed with insight, e.g. in relation to the importance of atrial remodeling in AF pathogenesis, including in the Nielsen study, on which a large part of this GWAS discovery is based (PMID: 30061737).
- d. “Heart contraction pathways play a pivotal role in ensuring the synchronized contraction of cardiac muscle fibers, which is essential for effective blood pumping” – It is difficult to understand the meaning of this sentence and how this knowledge relates to the results of the study. Furthermore, no references are cited.
- e. “However, given that AF is a multifactorial disease, our PRS with R² of 5.3% is still respectable for disease risk prediction.” - I cannot find this statistic anywhere in the results section (text or tables).

Response to comment 9: We thank the Reviewer for their careful evaluation of our work. Based on updated data and analyses, we have completely rewritten the discussion to reflect our new findings. For gene annotations at identified loci, we employed a comprehensive gene prioritization strategy, integrating six different approaches to enhance accuracy. Using the prioritized gene list, we conducted pathway enrichment analyses against the Reactome and GO databases, identifying both well-established and additional pathways potentially involved in AF development. Additionally, we updated our polygenic risk score (PGS) analysis, testing a PGS derived from our cross-ancestry GWAS in an independent cohort (Penn Medicine Biobank). We then compared its risk prediction performance with the PGS from Miyazawa et al., 2023. Our cross-ancestry-derived PGS demonstrated superior predictive accuracy, highlighting its significant improvement in disease risk stratification at the population level. For further details, please refer to our revised manuscript.

Reviewer #3:

Yuan et al. performed a large-scale meta-analysis of AF-GWAS including approximately 120,000 cases, identifying 178 loci associated with AF, 28 of which were previously unreported. Following this, genetic correlations between AF and relevant traits were assessed, polygenic risk scores were constructed, and potential risk factors and therapeutic targets were examined through proteome-wide Mendelian randomization analysis. Particularly in MR analysis, colocalization and sensitivity analyses were also performed to support the evidence of the MR results. The authors elaborated on the details in the analytical methods and the manuscript is well written, but it seems to be unclear along several issues and may need some additions.

Response to the comment: We sincerely thank the Reviewer for positive comments as well as constructive points. We have now carefully considered these comments and revised the paper accordingly.

Major comments;

1. When calculating PRS using the P+T method, multiple parameter values were tested, but for the PRS-CS method, no parameter settings were specified. Although this is not explicitly mentioned in the Methods section, was PRS calculated using the default value for the parameter such as phi in the PRS-CS? Since the P+T calculated PRS under several conditions, wouldn't it be worth trying to set the phi parameter under various conditions as well? Additionally, when performing such parameter tuning, I think it is essential to adjust the parameters using an independent validation cohort and then conduct performance evaluations with a separate test cohort.

Response to comment 1: Thank you for your comment. Given the inclusion of multi-ancestry data in our GWAS meta-analysis, we used PRS-CSx with the "auto" setting to construct the polygenic risk score (PGS), ensuring optimal ancestry-specific weighting. To evaluate the score, we applied it in an independent cohort (Penn Medicine Biobank), assessing its association with AF prevalence, odds ratio estimates, and overall risk prediction performance. We have now incorporated these details into the manuscript as suggested.

2. I think that deCODE proteome data is primarily used in the MR analysis. However, this is not mentioned in the Methods, even not in MR analysis in Supplementary Methods. This is only first mentioned in the colocalization analysis section in the Supplementary Methods. While it might be clear after reviewing all the materials, it would be better to introduce the data used in a more logical order, so that readers can understand it more easily, and provide the details in the supplementary materials.

Response to comment 2: Thank you for the point. We have updated our analytic strategy for the pQTL analysis to comprehensively identify causal proteins associated with AF (Figure 2). Specifically, we combined two pQTL datasets (deCODE and UKB-PPP) for instrument selection. After removing overlapping proteins, we included 2,847 proteins with cis-SNPs in the analysis. For proteins present in both datasets, we prioritized UKB-PPP data due to its larger sample size and the fact that Olink identified a greater number of cis pQTLs (Eldjarn et al., Nature, 2023). Notably, the associations with AF for these overlapping proteins demonstrated good consistency between the two datasets (See reply to the comment 6 of Reviewer #1). For overlapping protein panels between the deCODE and Fenland studies, we prioritized the Fenland study as the replication dataset because its smaller sample size resulted in fewer identified cis-pQTLs, making it more suitable for validating findings.

Regarding the statistical methodology, we transitioned from a cis two-sample MR analysis to SMR analysis (Zhu et al., Nature Genetics, 2016), which incorporates the HEIDI test to detect pleiotropy that could otherwise lead to false positives. This change was also prompted by comments from other Reviewers. To further strengthen causal inference, we implemented genetic colocalization analysis to rule out false positives driven by linkage disequilibrium. A stringent colocalization threshold ($PPH4 \geq 0.8$) was applied to ensure robust findings.

We have updated the corresponding sections on methods and results in the manuscript to reflect these changes and to emphasize the methodological rigor of our approach.

3. As the authors mentioned, sample overlap can introduce bias into MR results. While the sensitivity analyses were performed to ensure no sample overlap in the MR analysis, I think it would be better not to conduct MR analysis with sample overlap from the beginning and to only present the MR without sample overlap. Additionally, if the pQTL and AF-GWAS signals are strongest in the absence of sample overlap, it might be better to focus on the MR analysis using the Fenland study and the meta-analysis of AF-GWAS as the main analysis and use the MR with deCODE and other meta-GWAS excluding deCODE as a replicate.

Response to comment 3: We appreciate the Reviewer's concern regarding the issue of sample overlap in MR analyses. This challenge becomes unavoidable when leveraging large-scale GWAS meta-analysis data, as the UK Biobank is involved in many of these datasets. While large sample overlaps may introduce false positives due to model overfitting, the impact of this bias is minimized when the instrumental variable strength is robust, typically measured by an F statistic > 10 (Burgess S, et al., *Genet Epidemiol.* 2016). In our analysis, we calculated the F statistics for all traits and found them to be > 10 , suggesting that sample overlap bias is unlikely to substantially affect our results.

Regarding the pQTL MR analysis, we have revised our analytic strategy as detailed in the response to comment 2. Given the smaller sample size of the Fenland study, which results in fewer identified cis-pQTLs and the potential absence of associations between specific proteins and AF risk, we believe prioritizing deCODE as the discovery dataset is a more robust approach. Furthermore, we observed that the significant protein-AF associations identified in the discovery analysis were replicated using available genetic instruments from the Fenland study, with all associations showing P values < 0.05 . This consistency further supports the robustness and reliability of our findings.

4. Depending on the protein, multiple pQTLs may serve as instrument variables in MR analysis. Please provide details on the number of pQTLs for each protein used in the MR analysis. Additionally, when using multiple pQTLs of a given protein as instrumental variables, did the authors perform colocalization analysis between individual pQTLs and AF-GWAS and calculate the posterior probability? If so, did the authors consider the average or maximum posterior probability with more than 0.8 to have evidence of colocalization?

Response to comment 4: In the SMR analysis (see the response to comment 7a of Reviewer 2 for details), each protein was represented by a single lead cis-SNP as the instrumental variable. For the HEIDI analysis, multiple SNPs (ranging from 3 to 20) were used to assess potential pleiotropy. In the colocalization analysis, all SNPs within the cis region of the encoding gene for each protein were included. The number of SNPs included in the colocalization analyses are provided in the Table S11.

5. In the discussion section, the authors stated that muscle development and heart contraction were significant pathways in the gene-enrichment analysis and played critical roles. However, these two pathways have already been reported in the study by Patrick T. Ellinor et al. published in *Nature Genetics* in 2018. Therefore, it is necessary to clearly distinguish and

explicitly highlight the mechanisms revealed by the identification of new loci in this study from what has already been described in previous GWAS.

Response to comment 5: Thank you for the comment. We agree that muscle development and heart contraction are well-established etiological pathways for atrial fibrillation, as demonstrated in previous studies. Leveraging an expanded number of identified genetic loci and more precise gene prioritization, we re-performed pathway enrichment analyses using the Reactome and Gene Ontology (GO) databases. After Bonferroni correction, we identified 36 significant pathways in Reactome, along with 50 biological processes (BP), 7 cellular components (CC), and 6 molecular functions (MF) in GO. Notably, Reactome enrichment analysis highlighted “Drug-mediated Inhibition of CDK4/CDK6 Activity,” a pathway associated with an increased risk of atrial fibrillation, suggesting potential therapeutic relevance. Additionally, pathways such as “RUNX3 Regulates CDKN1A Transcription” and “Downregulation of TGF-beta Receptor Signaling” were identified, underscoring the critical role of the TGF-beta pathway in promoting AF through its contribution to atrial fibrosis.

GO enrichment analysis further validated the importance of muscle development and heart contraction in AF onset, while also revealing additional pathways, including the cellular response to transforming growth factor beta, artery morphogenesis, regulation of cell communication via electrical coupling, and actin filament-based movement, offering new mechanistic insights into AF. We have incorporated these results and the associated discussion into the text. While only the top pathways are displayed in Figure 3, detailed results from these analyses are provided in Table S5 and Table S6.

6. In the discussion section lines 324-326, the authors mentioned that lipids and diabetes may not be important risk whereas these factors are established ones in terms of clinical practice. If so, could you show more evidence to convince readers?

Response to comment 6: Thank you for the comment. We have re-conducted the MR analysis using an updated GWAS dataset that includes a larger number of cases, thereby enhancing the statistical power of the analysis. Our findings confirm that lipids and diabetes are associated with an increased risk of AF, aligning with current clinical practice and reinforcing the importance of managing these risk factors in AF prevention.

7. Figure 3. Enrichment analyses were performed using just the top 20 signals. Why 20? If you utilized all positive signals/ all negative signals/ all positive and negative signals, what result will we see?

Response to comment 7: Thank you for the comment. We have updated the enrichment analysis by adopting a more comprehensive and robust gene prioritization strategy. In this revised analysis, we utilized six complementary gene prioritization approaches: (1) Nearest Gene, (2) MAGMA, (3) PoPS, (4) eQTL mapping using colocalization analysis with eQTLGen, (5) encoding variants in credible sets identified by CARMA, and (6) TWAS. For each locus, the prioritized gene was determined by selecting the gene with the highest number of selections

across these six methods. In cases where multiple genes had equal counts, we prioritized genes encoding variants in credible sets identified by CARMA, followed by the nearest gene.

Using all prioritized genes across the genetic loci identified for AF, we performed two enrichment analyses based on Reactome and GO databases. After applying Bonferroni correction, we identified 36 significant pathways in Reactome, along with 50 biological processes (BP), 7 cellular components (CC), and 6 molecular functions (MF) in GO. The corresponding results are presented in Figure 3 and detailed further in the Table S5 and S6.

8. Figure 3. The author utilized GTEx data from the left ventricle and atrial appendage. While I agree that these two organs are associated with AF, how about other organs/ tissues? Stratified LDSC or DEPICT algorithm could point out additional AF-related ones.

Response to comment 8: We conducted both MAGMA tissue enrichment and S-LDSC analyses to identify tissues most associated with AF using data from GTEx V8 (53 tissues and cell types). Three tissues—Heart_Atrial_Appendage, Artery_Tibial, and Heart_Left_Ventricle—were identified as significantly associated with AF-related MAGMA genes. The S-LDSC analysis also showed the strongest enrichment in Heart_Left_Ventricle and Artery_Tibial. Thus, we included these three tissues for TWAS. Although Artery_Tibial is not directly a heart-related tissue, we included it due to its consistent association in both the MAGMA and S-LDSC analyses.

Furthermore, our pathway enrichment analysis was based on genes prioritized through a comprehensive set of six approaches. As such, tissue selection for TWAS is unlikely to substantially impact gene prioritization or the subsequent pathway enrichment results. WE have added these in the paper.

9. Figure 4. Now that the GWAS summary statistics of many traits are available, could you comprehensively investigate genetic correlations with other traits than traditional AF risk factors?

Response to comment 9: As suggested, we conducted a comprehensive genetic correlation analysis between AF and near 130 circulatory system outcomes using MVP data, separately in European and African American populations, to identify comorbidities between ancestries. After Bonferroni correction, we identified significant 95 genetic correlations in Europeans, compared to 18 in African Americans. Notably, the top correlations differed substantially between the two populations, suggesting ancestry-specific disease clusters associated with AF. We have added this in the paper.

10. Figure 5a. While the performance of PRSs was shown, is there statistical difference between those PRSs? You can test it using bootstrap method or just compare their AUROC using Delong test.

Response to comment 10: Thank you for the comment. We have now presented the 95% confidence interval for both PGS. Our PGS showed superior predictive performance compared to PGS002814 from the Miyazawa et al. study, with an area under a receiver operating characteristic (AUC) of 0.780 (95% CI: 0.778–0.783) and a Brier score of 0.092 (95% CI: 0.091–

0.093), outperforming PGS002814 (AUC = 0.767, 95% CI: 0.764–0.769; Brier score = 0.094, 95% CI: 0.093–0.095).

11. In the supplementary methods, it is stated the strong evidence of colocalization if the posterior probability for shared causal variants (PH4) was ≥ 0.8 , but in the main manuscript, it is described as having high support of colocalization with $PP.H4 > 0.7$. Which one is correct? Also, what is the rationale for choosing this threshold?

Response to comment 11: We apologize for the misunderstanding caused by typographical errors. In the revised version, we have consistently applied $PPH4 \geq 0.8$ as the threshold for strong colocalization evidence. A PPH4 value above 0.8 (or 80%) indicates that there is an 80% or greater probability that the same genetic variant is responsible for both the GWAS and pQTL signals. Using $PPH4 \geq 0.8$ as the colocalization threshold ensures high confidence that both traits share a common causal variant, minimizing the risk of false-positive colocalization due to linkage disequilibrium or independent associations. This threshold is also widely adopted in pQTL studies, reinforcing its validity in colocalization analyses.

Minor comments;

12. The figures cited in the Blood proteins and AF section are all labeled as Figure 6, but I think it should be Figure 7.

Response to comment 12: Checked as suggested. After the update, the results for the protein-AF MR analysis are now presented in Figure 6.

13. Regarding Figure 3c, the description of the figure legend is insufficient. There is no explanation for the abbreviations BB, CC, and MF, nor for the terms 'count' and 'gene ratio'.

Response to comment 13: Thank you for your comment. We have provided a detailed explanation of these concepts in the footnote of Figure 3.

14. Figure 5 is generally of low quality. For example, in Figure 5a, there are multiple bars of similar pseudo R^2 values, making it difficult to see under which condition the pseudo- R^2 is the highest. It would be helpful to reorder them in descending order. Additionally, the X-axis of Figure 5a represents PRS parameters and the number of SNPs, but the font size is small, making it difficult to understand. For the sake of clarity, it might be helpful to use color coding for methods like P+T and PRS-CS, or to display the parameters separately instead of including them as axis text. The label of the Y-axis in Figure 5c is missing. In Figure 5e, the labels for the significant phenotypes overlap and are hard to read. The phenotype categories are color-coded with two colors, but using separate colors for each category would make it easier for readers to understand.

Response to comment 14: Thank you for your comment. After updating the analysis, we have revised the results and figures and have also provided high-quality figures for clarity and accuracy.

Reviewer #2:

I thank the authors for their responses to my comments, and those of other reviewers. I have several remaining concerns, outlined below. In particular, there is a large overlap of cases between the current study and a recently published AF GWAS and the approach to claiming novel loci ignores published research (in addition to the newly published one). Furthermore, the authors claim differences in genetic correlation between ancestry groups without performing any statistical comparison. Finally, I stand by my previous criticism of claiming that proteins are causal in AF based on MR with single variant as instrument and not giving that variant and its AF association.

Response to the comment: We appreciate the Reviewers for assessing our work and providing insightful comments. We have improved the paper accordingly.

1. Overlap of cases with other AF GWASs

In my previous comments I pointed out that a majority of the discovery dataset was the exact same as the data of a 2018 GWAS meta-analysis of atrial fibrillation by Nielsen et al., i.e. the HUNT, deCODE, MGI, DiscoverEHR, AFGen consortium and UK Biobank datasets. Now the authors have updated UK Biobank (and now use it as a replication study) and added the publicly available MVP dataset to their study, which is by far the greatest addition to the European discovery dataset, as well as adding participants of other ancestries. Combined number of cases in the current study is now almost 170,000. However, since the first revision of the current paper, Roselli et al. have published the largest GWAS of AF hitherto, with over 180,000 cases (<https://pubmed.ncbi.nlm.nih.gov/40050429/>). The Roselli study includes updates of many of the cohorts for which the current study uses summary statistics from Nielsen et al., 2018, i.e. HUNT, deCODE, MGI, DiscoverEHR. In fact, it seems that the Simpler cohort of approximately 8500 individuals is the only cohort in the current study that is not included by Roselli et al. in the same or larger version. Otherwise, if I am not mistaken, the current study is largely a subset of the Roselli-2025 GWAS.

Response to comment 1: We acknowledge the reviewer's observation regarding sample overlap with prior GWAS meta-analyses. While our study shares a substantial portion of its discovery dataset with Roselli et al., it is important to note that the objectives and analytic focus of the two studies are distinct. Roselli et al. primarily aimed to identify novel genetic loci, improve the performance of polygenic risk scores, and examine comorbidities using an expanded GWAS meta-analysis of AF. In contrast, our study builds on this genetic foundation by (1) conducting population-specific analyses of circulatory comorbidities in both European and African populations, (2) integrating proteomic data to identify blood proteins associated with AF, and (3) developing a protein-based risk prediction score for AF. While we acknowledge the sample overlap and a slightly smaller total case number in our study compared to Roselli et al., our integrative multi-omics and population-informed approach addresses unique research questions and offers additional mechanistic and translational insights into AF risk beyond those explored in prior GWAS.

2. Novelty of identified loci

In light of the Roselli-2025 publication (<https://pubmed.ncbi.nlm.nih.gov/40050429/>), claims of novelty need to be revisited. Furthermore, the authors approach to defining novelty is incomplete: Loci without overlapping coordinates for previously reported genome-wide significant variants ($P < 5 \times 10^{-8}$) from prior studies (Nielsen JB et al., 2018;7 Roselli C et al., 2018;8 Miyazawa K et al., 2023) were annotated as novel.“ – it is unclear why the authors restrict to three studies when many more exist and a simple look-up in the GWAS catalog would improve definition of novel loci. Many loci are claimed as novel but have been reported in GWASs of AF (e.g. at BRWD1: <https://pubmed.ncbi.nlm.nih.gov/39024449/> and PLEC: <https://pmc.ncbi.nlm.nih.gov/articles/PMC5704994/>). Another recently published study by Choi et al (<https://www.nature.com/articles/s41588-025-02074-9>), reporting structural and coding variants in AF, would also need to be taken into account when claiming novelty.

Response to comment 2: Thank you for this important comment. In response, we have re-evaluated the novelty of our identified loci using a more comprehensive approach. Specifically, we performed a PheWAS of the lead variants in the Open Targets platform, which incorporates evidence from the GWAS Catalog, UK Biobank, and FinnGen. We then cross-referenced our findings with all studies mentioned by the reviewer—including the recent Roselli et al. 2025 publication, as well as prior GWASs by Thorolfsdottir RB et al 2017, Nielsen et al. 2018, Roselli et al. 2018, Miyazawa et al. 2023, Verma A et al 2024, Roselli et al. 2025, Choi et al. 2025, and other relevant literature. Based on this thorough comparison, we found that 237 of the 525 genome-wide significant loci identified in our analysis appear to be previously unreported in relation to atrial fibrillation. We have revised the relevant sections of the manuscript to clarify this updated definition of novelty and to reflect the enhanced rigor of our approach.

Page 17-18: *“Loci were annotated as unreported if loci had no overlapping coordinates with previously reported genome-wide significant variants ($P < 5 \times 10^{-8}$) associated with AF based on a comprehensive evaluation. This included PheWAS lookups using the Open Targets platform (<https://genetics.opentargets.org/>), integrating data from the GWAS Catalog, UK Biobank, and FinnGen), as well as cross-referencing with prior AF GWAS reports, including those by Thorolfsdottir et al. (2017),⁴⁹ Nielsen et al. (2018),⁷ Roselli et al. (2018, 2025),^{8,50} Miyazawa et al. (2023),⁹ Verma et al. (2024),⁴² Choi et al. (2025),⁵¹ and other relevant studies.”*

3. Publicly available data

The authors have not answered whether they had access to any individual level data (specifically I asked about Simpler). It would be appropriate to state somewhere in the manuscript if the whole study is based on publicly available summary statistics or not.

Response to comment 3: Thank you for the clarification request. We confirm that we had access to individual-level data on atrial fibrillation diagnosis and genotype information from the SIMPLER cohort. We have now clarified this in the manuscript.

Page 16: *“We performed GWAS association testing using individual-level genotype and phenotype data from participants in the SIMPLER cohort.”*

4. Gene prioritization

The authors have improved description of methodology for gene prioritization. However, it is questionable to put proximity above f.ex. eQTL and not to focus on eQTLs in appropriate tissues (i.e. the heart). Furthermore, many of the prioritized genes are only supported by proximity, which is very weak evidence.

Response to comment 4: Thank you for this valuable comment. We initially prioritized genes using eQTLGen due to its substantially larger sample size and higher statistical power, which improves sensitivity for detecting eQTLs. However, we fully agree that tissue specificity is essential, particularly for a cardiac trait such as atrial fibrillation. In response to your suggestion, we have now incorporated colocalization analyses using GTEx v8 eQTL data from heart atrial appendage and heart left ventricle tissues to enhance biological relevance. These analyses identified putative target genes at 146 loci (atrial appendage) and 141 loci (left ventricle). Detailed results have been presented in Table S4. While this additional layer of evidence improved tissue-contextual interpretation, it did not substantially alter the overall gene prioritization results. We have updated the manuscript accordingly to reflect this expanded approach.

We agree that proximity-based annotation provides limited functional insight on its own. However, prior benchmarking studies—including our own work (PMID: 37845353) and that of others (PMID: 39930082)—have shown that nearest-gene assignment can perform reasonably well in gene prioritization, particularly when other functional evidence is unavailable. That said, we acknowledge its limitations and have clearly indicated which prioritized genes are supported solely by proximity. Where possible, we have supplemented this approach with additional layers of evidence, including eQTL colocalization and fine-mapping data, to improve biological plausibility.

Page 14-15: *“Second, despite employing multiple prioritization strategies, some degree of gene misassignment is likely inevitable due to the limitations of current functional annotation resources. While many genes at these loci were prioritized based on proximity to the lead variant, we have explicitly noted when proximity was the sole criterion and, where possible, incorporated supporting evidence from eQTL colocalization and fine mapping to strengthen biological plausibility.”*

Page 20: *“For this analysis, we used eQTL data from eQTLGen Phase I and the Genotype-Tissue Expression (GTEx) Project v8 for heart atrial appendage and heart left ventricle tissues. Variants within 500 kb of each GWAS index variant were extracted to perform colocalization analysis.”*

5. Correcting for multiple testing in enrichment analysis

Regarding “Enrichment analysis in the Reactome database identified 36 out of 1,131 pathways significantly associated with AF after Bonferroni correction.” A bonferroni correction with 1,131 tests would give a p-value threshold of 4.4×10^{-5} . Please explain why 36 pathways are considered significant when only 5 are below this threshold according to Table S5.

Response to comment 5: We apologize for the oversight in our initial analysis, where we applied FDR correction instead of the more stringent Bonferroni correction for pathway enrichment in the Reactome database. After applying Bonferroni correction, five pathways remained statistically significant. We have updated the text, including revisions to the relevant text and Figure 3a to reflect this correction.

Page 6: *“Enrichment analysis in the Reactome database identified 5 out of 1,131 pathways significantly associated with AF after Bonferroni correction.”*

6a. Comparison of genetic correlation with CVDs between ancestry groups
Regarding “However, Europeans had additional associations with valve disease and cardiomegaly, while African Americans showed stronger links to rheumatic heart disease and cerebral ischemia.” – please provide an objective comparison of genetic correlation of AF and each of these phenotypes among Europeans and African Americans, e.g. is there significant heterogeneity?

Response to comment 6a: Thank you for this insightful comment. To more robustly identify traits with truly different genetic correlations between populations, we applied Cochran’s Q test to detect statistical heterogeneity in genetic correlation estimates between European and African populations. We defined population-specific differences as those traits with $I^2 > 75\%$ and P value for Cochran’s Q < 0.05 .

Following the updated heterogeneity assessment, we identified several phenotypes for which the genetic correlation with AF may differ between Europeans and Africans. We have incorporated the updated heterogeneity analysis and the corresponding interpretation into the manuscript.

Page 7: *“Among the traits assessed for heterogeneity in genetic correlation with AF between European and African populations, several phenotypes demonstrated substantial population-specific differences. We identified conditions such as first-degree atrioventricular block, abdominal aortic aneurysm, varicose vein of lower extremity, deep vein thrombosis, tachycardia, transient cerebral ischemia and abnormal heart sounds as having significantly heterogeneous genetic correlations with AF across ancestries (Figure 4).”*

Page 22: *“To objectively compare genetic correlations between populations, we applied Cochran’s Q test to assess heterogeneity in the correlation estimates between European and African populations. Traits were considered to exhibit population-specific differences if they showed evidence of substantial heterogeneity, defined as an I^2 statistic greater than 75% and a Cochran’s Q test P value less than 0.05.”*

6b. Furthermore, the authors cite two studies on the matter: “We observed ancestry-specific differences in genetic correlations between AF and circulatory comorbidities, with Europeans showing stronger associations with valve disease and cardiomegaly, while African Americans had stronger links to rheumatic heart disease and cerebral ischemia. These disparities, also reported in other studies,^{19,20} may arise from ancestry specific genetic architecture,

environmental factors, and variations in AF subtypes” – the studies that are cited are observational, not genetic studies, which would be relevant to mention. Furthermore, these studies do not mention differences in valve disease, cardiomegaly and rheumatic heart disease, as the text indicates. One of them (Magnani et al.) examines racial differences in the outcome of AF on the rates of stroke, heart failure, CHD, and mortality and the other (Essien et al.) examines racial differences in anticoagulation prescription and stroke.

Response to comment 6a: Thank you for this thoughtful comment. Following our updated heterogeneity assessment, we identified several circulatory phenotypes—such as first-degree atrioventricular block, abdominal aortic aneurysm, varicose veins of the lower extremities, deep vein thrombosis, tachycardia, transient cerebral ischemia, and abnormal heart sounds—that demonstrated significantly different genetic correlations with AF between European and African populations. Given the exploratory nature of these analyses and the limited availability of prior genetic studies directly comparing population-specific genetic correlations for these traits, we have removed the earlier discussion referencing specific comorbidity differences and citations that were not directly relevant. Instead, we now acknowledge the limitations of this finding and discuss it in broader terms, emphasizing the need for further research in diverse populations to validate and better understand these population-specific patterns.

Page 11-12: “We observed significant heterogeneity in the genetic correlation between AF and several circulatory phenotypes across European and African populations. While these findings suggest the possibility of population-specific differences in the shared genetic architecture between AF and its comorbidities, we acknowledge that these results are exploratory and require validation in independent cohorts. Due to the limited number of prior genetic studies addressing population-specific correlations for these traits, we refrain from drawing strong conclusions about the direction or clinical implications of individual trait differences. Instead, our findings underscore the broader need for population-informed genetic analyses and increased representation of diverse populations. Facilitating this type of research may improve the accuracy of risk stratification, inform targeted screening strategies, and reduce disparities in cardiovascular outcomes across diverse patient populations”

7. Mendelian randomization – correcting for multiple testing

Regarding “Additionally, genetically proxied high-density lipoprotein (HDL) cholesterol levels, cigarettes smoked per day, moderate-to-vigorous physical activity, sleep duration, short sleep time, and blood calcium levels were associated with AF at the nominal significance level.” - would recommend adhering to claiming only associations that are significant after correcting for multiple testing.

Response to comment 7: We have removed these sentences and corresponding discussion in the paper as suggested.

8. MR-analysis using cis-pQTLs as instruments to identify causal proteins

In my first review I asked if only one instrument was used for each protein-AF MR analysis. The authors have responded that only one cis-pQTL is used in each case. However, they have not

responded to my request to give information on which snp is being used as instrument. Knowing the SNP and its association (OR and P-value) with AF is crucial – are the AF associations at noise level considering the large sample size for AF? And if these proteins are causal in AF, would the cis-pQTLs not come up in the AF GWAS? Furthermore, for three of the identified “causal” proteins, effects on AF in the replication dataset (Fenland) were opposite to that of the discovery – could this indicate that the instruments used are not true AF associations?

Response to comment 8: Thank you for raising this important point. We have now included information on the genetic instruments (cis-pQTLs) used for each protein, along with their association statistics with AF, in the Table S20 as requested.

While it is reasonable to hypothesize that cis-pQTLs for causal proteins might also show associations in a GWAS for atrial fibrillation (AF), the absence of genome-wide significant associations for these variants does not undermine the validity of the MR findings. GWAS must correct for a large multiple testing burden—typically across millions of variants—which limits power to detect associations with modest effect sizes, especially for variants whose influence on disease risk is mediated through intermediate molecular traits such as protein levels. In contrast, MR leverages prior biological knowledge to test a restricted set of variants with known effects on a specific exposure, which substantially reduces the multiple testing burden and increases the statistical power to detect causal relationships. Moreover, pQTLs are identified based on associations with continuous traits (protein abundance), which inherently provide greater statistical power compared to case-control GWAS designs, assuming similar sample sizes and variance explained. In addition, although the sample size and number of AF cases in the GWAS do influence MR power, it is important to emphasize that MR relies primarily on the strength of the variant’s association with the exposure (i.e., protein levels) rather than a strong association with the outcome. A variant may robustly influence protein concentration yet exhibit no detectable signal in the AF GWAS if its effect on disease risk is indirect, attenuated, or context dependent. Such variants may also evade detection due to modest downstream effect sizes or clinical heterogeneity in the AF phenotype. Therefore, the lack of genome-wide significance in the outcome GWAS does not preclude a genuine causal role as inferred from well-powered, biologically anchored MR analyses.

Regarding the directionally discordant effects observed in the replication (Fenland) dataset for three proteins, we note that this occurred in a minority of cases. These discrepancies may reflect several factors beyond invalid instruments, including: (1) differences in genetic effects on protein levels across populations; (2) variability introduced by different proteomic platforms; (3) measurement error or batch effects; and (4) chance, given the reduced sample size and power in replication settings. While such inconsistencies warrant caution, they do not necessarily invalidate the MR findings, particularly when the discovery results are robust and biologically plausible. Overall, we interpret these findings as hypothesis-generating and acknowledge the need for further validation in independent datasets and mechanistic studies.

Page 13: *“Nonetheless, we observed that a subset of associations could not be replicated in the independent dataset. While such discrepancies may arise from differences in genetic regulation across populations, platform-specific variation in protein quantification, or measurement error in replication analyses, they do not necessarily invalidate the MR results. However, they do*

warrant caution in interpreting these findings. Importantly, the associations identified in our study are based on protein levels measured in circulation and may not fully capture tissue-specific effects relevant to AF pathogenesis. Further validation in independent cohorts and functional characterization of these proteins in cardiac-relevant tissues and models will be essential to confirm their causal roles and assess their translational potential.”

Page 24: “Detailed information on selected genetic IVs is shown in Table S20.”

9. Discussion:

a. “The PITX2 and ZFH3 have been recognized as well-known genes associated with AF, our study emphasized these two as shared loci across four ancestry groups and thus further confirms their pivotal role in AF susceptibility.” - this is not novel, these loci have been well established in different ancestries through other studies (e.g. <https://pmc.ncbi.nlm.nih.gov/articles/PMC6136836/> and <https://pmc.ncbi.nlm.nih.gov/articles/PMC9925380/>)

Response to comment 9a: Thank you for this important observation. We agree that the association of PITX2 and ZFH3 with AF has been well established across multiple ancestries in prior studies. We have revised the sentence to acknowledge this and now present our findings as a replication and reinforcement of their consistent association with AF across diverse populations, rather than a novel discovery.

Page 11: “PITX2 and ZFH3 are well-established AF-associated genes; our findings reaffirm their consistent association across four population groups,8,9 further supporting their pivotal role in AF susceptibility.”

b. Two paragraphs are almost exact replicas of each other, starting with:
“We employed a comprehensive gene prioritization strategy, identifying putative causal genes for 504 loci, providing functional insights into AF pathogenesis...”
“We used a series of methods for gene prioritization and identified putative causal genes for 504 loci, providing functional insights into AF pathogenesis...”

Response to comment 9b: Thank you for pointing this out. We apologize for the redundancy and have removed the second, duplicative paragraph in the revised manuscript to improve clarity and readability.

Reviewer #3:

The authors have responded to my comments appropriately. Just one thing, 95% CI has been added regarding the performance of PRS, making the numerical difference easier to understand. However, it is still unclear whether the performance has improved statistically significantly.

Response to the comment: We sincerely thank the Reviewer for the positive feedback. To address the remaining concern, we conducted the DeLong test to formally assess whether the difference in predictive performance between the two polygenic scores is statistically significant. The test compares the area under the ROC curves (AUC) for the two models. Our analysis showed that the AUC of the PGS derived from the GWAS meta-analysis was significantly higher than that of the Miyazawa PGS (p value $< 2.2e-16$). This supports that the improvement in predictive performance is not only numerically greater but also statistically significant. We have added the DeLong test results and corresponding interpretation to the Results section.

Page 9: *“The DeLong test showed that the AUC of the PGS derived from our GWAS meta-analysis was significantly higher than that of the Miyazawa PGS ($P < 2.2e-16$).”*

Page 26: *“We used the DeLong test, implemented in the pROC R package, to statistically compare the AUROC of the polygenic scores and evaluate whether the difference in predictive performance was significant.”*

Reviewer #4:

This is my initial exposure to the current manuscript as I was asked to review the revision on behalf of a prior reviewer. Therefore, my comments are focused on whether the prior reviewer’s concerns were adequately addressed. In general, the revisions have been comprehensive and additive, and the work is a worthy addition to the literature. I do note a few remaining issues:

Response to the comment: We sincerely thank the Reviewer for assessing our paper and providing constrictive comments. We have carefully considered these points and revised the paper accordingly.

1. I note that a large cross-ancestry AF GWAS meta-analysis was recently published in Nature Genetics (PMID 40050429). The current study has actually somewhat fewer cases (168k vs 180k), although more loci are identified. Referencing this recent paper would require a major overhaul to the paper and text, so I will defer to the Editors and journal policy as to whether any action needs to be taken to this effect.

Response to comment 1: We thank the Reviewer for this point. Along with comments from Reviewer 2, we have improved the strategy for unreported locus definition. We have first conducted a PheWAS of identified loci in Open Targets Genetics reveal whether these variants have been previously associated with atrial fibrillation. After this, we compared our results with that from large-scale AF GWASs including the recent AF GWAS to further define unreported loci. We have updated this part of results in the paper.

Page 5: *“Of the 525 genome-wide significant loci, 237 were unreported.”*

Page 17-18: *“Loci were annotated as unreported if loci had no overlapping coordinates with previously reported genome-wide significant variants ($P < 5 \times 10^{-8}$) associated with AF based on a comprehensive evaluation. This included PheWAS lookups using the Open Targets platform (<https://genetics.opentargets.org/>), integrating data from the GWAS Catalog, UK Biobank, and FinnGen), as well as cross-referencing with prior AF GWAS reports, including those by Thorolfsson et al. (2017),⁴⁹ Nielsen et al. (2018),⁷ Roselli et al. (2018, 2025),^{8,50} Miyazawa et al. (2023),⁹ Verma et al. (2024),⁴² Choi et al. (2025),⁵¹ and other relevant studies.”*

2. The ‘replication’ design here seems somewhat of an afterthought, as the replication dataset is then simply incorporated into the European and cross-ancestry meta-analyses which are (I think reasonably) presented as the primary analysis. The authors should consider whether the replication design really adds anything here, or whether streamlining the analysis would increase clarity.

Response to comment 2: Thank you for this insightful comment. We agree that the replication component may appear secondary in structure. However, our intent was to ensure that we did not miss genetic loci that might reach genome-wide significance only after meta-analyzing the discovery and replication datasets. The UK Biobank, used as the replication dataset, contributes a substantial number of AF cases—approximately one third of the total in the discovery phase—and therefore offers considerable power. Indeed, several loci were identified only after combining both datasets. We have clarified this rationale in the revised manuscript to better reflect the contribution of the replication dataset to the overall discovery effort.

Page 17: *“Genetic loci in the European analysis were defined based on a GWAS meta-analysis that combined both the discovery and replication datasets. This integrated approach maximized statistical power, enabling the identification of several loci that reached genome-wide significance only after the datasets were meta-analyzed.”*

3. The authors do not appear to apply a MAF filter (I just see $MAC < 50$). Note this is in contrast to the other large AF meta-analyses this study is compared to. This issue should be clarified, and the number/% of significant variants with $MAF < 1\%$ should be reported. Were more stringent QC parameters applied to rarer variants? Did rarer variants replicate (if the replication approach is retained)?

Response to comment 3: Thank you for this helpful comment. While we applied a $MAC < 50$ filter rather than a direct $MAF < 1\%$ threshold to exclude rare variants, we acknowledge the importance of consistency with prior AF meta-analyses. In practice, this distinction had minimal impact: only 5 genome-wide significant variants in the European meta-analysis and 7 in the cross-population meta-analysis had $MAF < 1\%$. Among the European-specific variants, 2 of the 5 low-frequency associations were replicated. For 7 loci in cross-population GWAS meta-analysis, 3 loci have been reported to be associated with AF in other studies. Given the small number of such variants, we believe the use of a MAC-based filter does not materially affect comparability with earlier studies. As suggested, we have now clarified this point in the revised manuscript.

Page 14-15: *“Third, although we applied a MAC < 50 threshold to exclude rare variants, a small number of variants with minor allele frequency (MAF) < 1% remained in the analysis (5 out of 493 in European GWAS and 7 out of 525 in cross-population GWAS). However, nearly half of these variants were replicated in our independent replication dataset or have been previously reported in association with AF in other studies. Given their limited number and supporting evidence, we believe that the inclusion of these variants does not materially affect the comparability of our results with earlier GWAS.”*

4. The gene prioritization approach appears substantially more robust and systematic in this version compared to the original. However, I feel the relevant language (Results: “we identified a single putative causal gene at each of the 504 genome-wide significant loci”, Limitations: “we cannot entirely rule out the possibility of misassigning genes to loci”) is too strong. The authors employ a robust approach but functional annotation remains imperfect and I would argue that misassignment of at least some genes is almost certainly present and inevitable. I recommend the authors nuance their reporting accordingly. To this end, it would be useful if the authors could report the number (%) of loci at which all available prioritization schemes prioritized the same gene.

Response to comment 4: We appreciate the reviewer’s positive feedback on the improvements to our gene prioritization strategy and fully agree with the suggestion to temper the language used in the manuscript. While we employed a robust and systematic framework to prioritize likely causal genes, we acknowledge that functional annotation remains imperfect and that misassignment is almost certainly unavoidable at some loci. We have revised the relevant statements in both the Results and Limitations sections to reflect this uncertainty more appropriately.

We also quantified the degree of concordance across gene prioritization approaches by reporting the proportion of genes that were consistently prioritized by all available methods. Specifically, we found that 47% of prioritized genes had $\geq 80\%$ agreement across methods. This level of agreement provides reassurance that our prioritization strategy is reasonably reliable, even though some misassignment is likely inevitable due to limitations in current functional annotation resources. Integrating multiple complementary approaches increases the robustness of gene prioritization, but we recognize that additional validation—particularly through experimental or tissue-specific data—remains essential.

Figure. The proportion (Y axis) of genes that were consistently prioritized by all available methods against the number of genes (X axis).

Page 6: *“Using a systematic prioritization framework, we nominated a likely causal gene at each of the 504 genome-wide significant loci, acknowledging that this assignment is based on available functional evidence and may not be definitive for all loci. Among these, 70 genes harbored protein-altering variants and 47% of prioritized genes had $\geq 80\%$ agreement across available methods (Table S4).”*

Page 14: *“Second, despite employing multiple prioritization strategies, some degree of gene misassignment is likely inevitable due to the limitations of current functional annotation resources. While many genes at these loci were prioritized based on proximity to the lead variant, we have explicitly noted when proximity was the sole criterion and, where possible, incorporated supporting evidence from eQTL colocalization and fine mapping to strengthen biological plausibility.”*

5. It would be helpful to state in the Results that the PGS analysis in PMBB represents an evaluation in an independent dataset not used for PGS derivation. The authors should also report the ancestral breakdown of the PMBB dataset used for evaluation.

Response to comment 5: Thank you for this helpful suggestion. We have now explicitly stated in the Results section that the PGS evaluation in the Penn Medicine BioBank (PMBB) was conducted in an independent dataset that was not used for PGS derivation. Additionally, we have added the ancestral breakdown of the PMBB participants included in the analysis to provide further context regarding the generalizability of our findings.

Page 9: *“To evaluate the performance of the polygenic risk score in an independent dataset, we tested it in the Penn Medicine BioBank (PMBB), which is not used for PGS derivation.”*

Page 26: *“The population breakdown of PMBB participants is predominantly European (~70%), followed by African (~25%), with smaller proportions of South Asian, East Asian, Admixed American, and other ancestries (Figure S20).”*

6. Regarding the protein scores, I do not fully understand why 2 approaches are used (LGBM + LASSO). It's specifically not clear to me the need for the LGBM, which is subjected to several steps prone to overfitting (e.g., ranking based on individual information gain, clustering, sequential forward selection), and unsurprisingly is outperformed by the LASSO. This is another potential area where there is unnecessary complexity. I also feel it is not helpful (for either model) to present results from the training set.

Response to comment 6: Thank you for this valuable comment. We agree that the inclusion of both LGBM and LASSO introduced unnecessary complexity and may have detracted from the clarity of the analysis. In line with the reviewer's suggestion, we have removed the LGBM-based approach from the revised manuscript. We now focus solely on the LASSO model, which offers a more interpretable and robust framework for protein score development. Additionally, we have restricted performance reporting to the testing set only to avoid potential bias and overfitting concerns associated with training set results.

7. The association between genetic liability to AF and lower levels of NTproBNP is potentially surprising and merits further discussion. I suppose the current statement that 'NTproBNP elevation in AF is likely a secondary effect of atrial stretch rather than a primary causal factor' could follow from this observation, it seems like an overly strong and somewhat simplistic interpretation, particularly as other factors (e.g., body mass index) may be at play.

Response to comment 7: We thank the Reviewer for this thoughtful comment. We agree that the association between genetic liability to AF and increased levels of NT-proBNP warrants careful consideration. Our findings are consistent with several prior observational studies reporting elevated NT-proBNP concentrations in individuals with AF compared to those without (PMID: 34850596; 35112889). In response to this comment and to further investigate the relationship, we conducted MR analyses of several established AF risk factors in relation to NT-proBNP (see response to comment 8). We found that genetically predicted waist-to-hip ratio (WHR), a key body composition trait, was inversely associated with NT-proBNP levels—consistent with earlier findings (PMID: 35851710; 33754794). However, reverse MR analyses revealed no significant association between genetic liability to AF and WHR, suggesting that body composition is unlikely to confound the observed AF–NT-proBNP association via horizontal pleiotropy. Additionally, recent studies have reported that higher NT-proBNP levels may predict incident AF (PMID: 39643423), highlighting the bidirectional and potentially complex nature of this relationship. Taken together, these findings suggest that our original statement may have oversimplified this interplay. We have therefore revised the manuscript to present a more nuanced interpretation of the association between AF and NT-proBNP.

Page 13: *"Genetic liability to AF was associated with increased levels of NT-proBNP, consistent with several observational studies that have reported elevated NT-proBNP concentrations among individuals with AF compared to those without.^{36,37} While this finding may reflect a downstream physiological effect of AF, such as atrial stretch or hemodynamic stress, the relationship is likely more complex. Recent studies have also shown that higher NT-proBNP levels may be associated with incident AF,³⁸ suggesting the potential for bidirectional influence. Taken*

together, even though our MR finding supports the interpretation that NT-proBNP elevation in AF is unlikely to reflect a primary causal mechanism, we should interpret this association with caution and emphasize the need for further investigation to disentangle the temporal and biological pathways linking NT-proBNP and AF."

8. Related to point #7 above it would be of interest to perform reverse MR for AF to "risk factors". I cannot tell whether this is an analysis that was in the prior version of this manuscript but removed due to prior author comments. If the latter is the case, then this comment may be ignored at the authors' and editors' discretion.

Response to comment 8: We thank the Reviewer for this constructive suggestion. We believe this comment was raised in the context of evaluating potential pleiotropic effects that could bias the observed association between genetic liability to AF and circulating NT-proBNP levels. To address this, we conducted an MR analysis assessing a wide range of established AF risk factors in relation to NT-proBNP. Interestingly, we found that several genetically proxied risk factors for AF were inversely associated with NT-proBNP levels (Table 1).

For example, waist-to-hip ratio (WHR), a proxy for central obesity, showed a significant inverse association with NT-proBNP (Table 2), consistent with prior reports (PMID: 35851710; 33754794). We also tested the reverse association—genetically proxied AF in relation to WHR—and observed a nonsignificant positive effect. Based on this and a directed acyclic graph (Figure 1) framework, WHR appears unlikely to confound or mediate the observed AF–NT-proBNP relationship via pleiotropy. These results support the interpretation that the association between AF and NT-proBNP is unlikely to be fully explained by pleiotropic effects from common AF risk factors.

Exposure	Outcome	nsnp	Beta	SE	P
Apolipoprotein A-I	NTproBNP	382	0.043	0.016	0.006
Apolipoprotein B	NTproBNP	228	-0.013	0.016	0.433
LDLC	NTproBNP	202	-0.003	0.016	0.866
Type 2 diabetes	NTproBNP	491	-0.024	0.008	0.004
Body mass index	NTproBNP	308	0.007	0.022	0.767
Waist-to-hip ratio	NTproBNP	567	-0.137	0.021	2.30E-10
Visceral adiposity	NTproBNP	292	0.027	0.020	0.189
Systolic blood pressure	NTproBNP	226	0.059	0.026	0.024
Diastolic blood pressure	NTproBNP	271	0.050	0.041	0.218
Thyroid-stimulating hormone	NTproBNP	43	-0.042	0.015	0.007
Smoking initiation	NTproBNP	312	0.019	0.019	0.310
Lifetime smoking index	NTproBNP	126	0.055	0.039	0.155
Leisure screen time	NTproBNP	131	-0.025	0.025	0.319
Insomnia	NTproBNP	205	-0.018	0.011	0.107

Table 1. Genetically proxied risk factors for AF in relation to levels of NTproBNP using data from UKB-PPP.

Outcome	nsnp	Method	Beta	SE	P
WHR	390	IVW-fixed	0.005	0.002	0.031
		IVW-random	0.005	0.006	0.395
		Weighted medium	0.005	0.005	0.258
		MR-Egger	-0.001	0.011	0.934
		MR-PRESSO	0.003	0.004	0.441

Table 2. Genetically proxied AF in relation to levels of waist-to-hip ratio.

Figure 1. The directed acyclic graph illustrating relationship genetic instruments, AF, obesity, and NTproBNP.

Reviewer #2:

I thank the authors for their responses to my comments. I want to respond further to their answer to comment #8 (protein-AF MR analysis):

I thank the authors for providing information on the instruments used in pQTL-MR analysis and their association with AF (although giving the protein name in ST20 would help, instead of just the protein ID). I argue that the association with the outcome is relevant. The AF meta-analysis is highly powered so associations close to nominal significance are scattered throughout the genome. I also want to point out that although the instrument's association with the exposure is well established, the exposure/outcome relationship tested is not based on biological plausibility. Thus, the authors are not testing a restricted set of biologically plausible causal factors, they are in fact performing a hypothesis free search for significant MR-analyses using 2,847 cis-pQTLs as instruments, and provide no reference to biological plausibility of a causal relationship between the proteins and AF. Therefore, it could be argued that it would be appropriate to apply a P-value threshold corrected for multiple testing to claim significance for the pQTL-MR analyses.

Response to the comment: We thank the Reviewer for raising this point. We have now added the full list of protein names in Supplementary Table 20. In addition, we apologize for the omission in the Methods: we did indeed apply a Bonferroni correction to our SMR protein analyses to control for multiple testing, and this has now been explicitly described in the revised Methods section.

Page 25: *“Bonferroni correction was used for multiple testing for SMR analysis.”*

Reviewer #4:

Most of my comments have been well-addressed. Just one remaining issue on my end.

I still think the NTproBNP MR findings are not sufficiently handled. Looking at ST15 (UKB) there is an association between liability to AF and lower levels of NTproBNP as well as lower levels of NPPB (NTproBNP precursor). In ST14 (deCODE), there is an association between liability to AF and higher levels of NPPB (and interestingly no representation of NTproBNP at all). By the way, in ST14 there are two entries for NPPB (rows 4 and 13), with different effect sizes, both positive, and I'm not sure what to make of that.

The association of liability to AF and lower levels of NTproBNP is correctly cited in the Results, but in the Discussion the authors seem to write the opposite: “genetic liability to AF was associated with increased levels of NTproBNP, consistent with several observational studies.” I believe there is some confusion here. The authors should revise the discussion to align with the results, which are not immediately consistent with the observational findings (which is what prompted my comment in the first place). Further, the conflicting associations for NPPB are also probably worth mentioning. In sum though, I

do generally agree with the authors interpretation that the evidence here does not support a causal role for AF on observed NTproBNP elevations in AF, but that relations are likely complex, potentially bidirectional, and require further investigation.

Response to the comment: We thank the Reviewer for highlighting this inconsistency and for the suggestion to mention the divergent NPPB findings. Two NPPBs in ST14 with different SOMA IDs are likely to be isomers. We have carefully revised the Discussion so that it now correctly reflects our MR result—that genetic liability to AF is associated with decreased NT-proBNP levels. We have also revised the corresponding discussion part to acknowledge the inconsistent associations between AF genetic risk and NPPB across two independent proteomic datasets. Finally, we expanded the text to note that, although elevated NT-proBNP predicts incident AF in cohort studies, our SMR analyses did not support a causal effect of genetically predicted NT-proBNP or NPPB on AF risk.

Page 13: *“MR revealed that genetic liability to AF is paradoxically associated with lower circulating NT-proBNP levels, in direct contrast to case-control studies reporting elevated NT-proBNP among AF patients.^{36,37} This discordance implies that the NT-proBNP elevations seen in AF may largely reflect secondary hemodynamic stress and atrial stretch rather than a primary effect of AF itself. Moreover, we observed inconsistent associations between genetic liability to AF and NPPB—the prohormone precursor to NT-proBNP—across two independent proteomic datasets, underscoring additional complexity. Although longitudinal cohorts have linked higher baseline NT-proBNP to subsequent AF,³⁸ our SMR analyses did not support a causal influence of genetically proxied NT-proBNP or NPPB on AF risk. Together, these data argue against a simple, unidirectional causal relationship between AF and NT-proBNP, and highlight the need for detailed longitudinal and mechanistic studies to untangle cause from consequence in the AF–NT-proBNP axis.”*